



# ChinaAI-FSC: A Comprehensive AI-Ready MODIS Fractional Snow Cover Dataset for China (2000-2022)

Jinliang Hou[1], Mingkai Zhang[1,2], Xiaohua Hao[1], Jifu Guo[3], Peng Dou[1], Ying Zhang[1*], Chunlin Huang[1,4*]

[1] Heihe Remote Sensing Experimental Research Station, State Key Laboratory of Cryospheric Science and Frozen Soil Engineering, Northwest Institute of Eco-Environment and Resources, Chinese Academy of Sciences, Lanzhou 730000, China
[2] University of Chinese Academy of Sciences, Beijing, 100094, China
[3] College of Information Science and Technology, Gansu Agricultural University, Lanzhou 730070, China
[4] Faculty of Geomatics, Lanzhou Jiaotong University, Lanzhou 730070, China

*Correspondence to*: Ying Zhang (zhang_y@lzb.ac.cn), Chunlin Huang (huangcl@lzb.ac.cn)

**Abstract.** We present ChinaAI-FSC, the first large-scale, standardized, AI-ready fractional snow cover (FSC) sample collection for mainland China, spanning 22 snow seasons from 2000 to 2022 and addressing a critical gap in long-term snow monitoring. The dataset consists of 47,728 samples (each 128 × 128 MODIS-pixel tiles), where high-resolution Landsat-5/7/8/9 and Sentinel-2 imagery provide consistent FSC reference labels. A total of 20 feature variables, including MODIS surface reflectance (bands 1-7), topographic attributes, forest and land cover information, and geolocation factors, were extracted to enable both point-scale and tile-scale spatially contextualized AI modelling. A structured and transparent workflow, encompassing systematic sample preparation, rigorous quality control, spatiotemporal sample partitioning, and standardized metadata, ensures reproducibility, physical consistency, and interoperability across machine learning and deep learning applications. Dataset reliability and AI-readiness were systematically evaluated using a novel "Four Layers-Four Domains-Fifteen Attributes (4L-4D-15A)" assessment protocol, covering data, information, system, and application dimensions. The quality, reliability, and usability of ChinaAI-FSC were demonstrated through three representative use cases: (1) benchmarking of six ML/DL models (ANN, SVR, RF, CNN, UNet, and ResNet), (2) validation of the standard MODIS FSC product, and (3) nationwide seamless FSC mapping. By providing harmonized, validated, and well-documented samples, ChinaAI-FSC establishes a unified foundation for AI-driven snow cover mapping, long-term monitoring, and cryosphere–hydrological modelling, promoting reproducible, interoperable, and next-generation research in cryospheric science. The dataset is publicly available from the National Tibetan Plateau Data Center (TPDC) at https://doi.org/10.11888/Cryos.tpdc.303034 (also accessible via https://cstr.cn/18406.11.Cryos.tpdc.303034) and from Zenodo at https://doi.org/10.5281/zenodo.17707386.
Key words: Fractional Snow Cover (FSC); AI-Ready; ML/DL; MODIS

## 1 Background and Motivation

Fractional Snow Cover (FSC) is a fundamental indicator for monitoring snowpack dynamics, as it quantifies the proportion of snow within a pixel, providing a continuous measure of snow extent that goes beyond binary snow/no-snow classifications.



Scientifically, FSC is a critical variable linking snowpack dynamics with energy and water exchanges at the land-atmosphere interface. It strongly influences surface albedo and shortwave radiation absorption, thereby affecting the timing of snowmelt, soil moisture evolution, and local to regional atmospheric circulation (Hall & Riggs, 2007; Frei et al., 2012). Accurate FSC

information enhances the representation of snow processes in land surface and climate models, helping to reduce biases in surface energy budgets and improve projections of climate feedback (Thackeray & Fletcher, 2016; Mudryk et al., 2020). From an applied perspective, FSC is indispensable for hydrological forecasting and water resource management, as it governs meltwater contributions to rivers and reservoirs, influencing agricultural planning, flood risk assessment, and hydropower operations (Barnett et al., 2005). Moreover, Furthermore, long-term, high-accuracy FSC records are essential for detecting

cryospheric responses to climate change, guiding adaptation strategies, and supporting international climate assessments such as those conducted by the IPCC. Despite its importance, operational estimation of FSC continues to face challenges arising from cloud contamination, complex terrain, vegetation canopy effects, and sensor limitations (Salomonson & Appel, 2004; Stillinger et al., 2023).

Early approaches to FSC retrieval primarily relied on statistical regression techniques, such as linear and exponential models,

as well as spectral mixture analysis (Salomonson & Appel, 2004, 2006; Painter et al., 2009). Statistical regression models establish empirical relationships between FSC and a limited set of spectral band combinations. Although computationally straightforward, their performance is highly sensitive to regional and seasonal variability, often resulting in poor generalization and systematic biases under heterogeneous surface conditions (Hall et al., 2002; Raleigh et al., 2013; Xin et al., 2012). Spectral mixture models, in contrast, assume that pixel reflectance is a linear or nonlinear combination of pure endmembers, which can

partially alleviate mixed-pixel effects. However, their accuracy heavily depends on the quality and representativeness of the endmember library, while the endmembers themselves can vary substantially with snow conditions and land-cover types. These limitations constrain the applicability of spectral mixture analysis in complex terrain and spatially heterogeneous environments (Dozier et al., 2008; Metsämäki et al., 2012; Painter et al., 2003, 2009; Rittger et al., 2013).

In recent years, FSC estimation has evolved from traditional empirical regression and spectral mixture decomposition methods

toward data-driven machine learning (ML) and deep learning (DL) paradigms. Early studies applied artificial neural networks (ANNs) that integrated MODIS surface reflectance with auxiliary variables to improve FSC estimation (Dobreva & Klein, 2011; Hou & Huang, 2014). Czyzowska-Wisniewski et al. (2015) further demonstrated the capability of ANNs in mountainous forests, highlighting their advantage in modelling nonlinear interactions among vegetation, terrain, and snow cover. Kuter et al. (2018, 2021, 2022) compared multiple ML algorithms, including multivariate adaptive regression splines (MARS), random

forest (RF), and support vector regression (SVR), revealing the respective strengths of these approaches for FSC modelling. With advances in big data and high-performance computing, research on FSC mapping has increasingly shifted toward DL-based multisource spatiotemporal fusion frameworks, especially in complex plateau and mountainous regions where snow distributions exhibit pronounced spatiotemporal heterogeneity. In this context, Azizi et al. (2024) and Liu et al. (2024) proposed convolutional neural network (CNN)-attention hybrid architectures (FSC-USNet) to effectively capture spatiotemporally

heterogeneous snow distributions in complex mountainous terrains. Similarly, Zhao et al. (2024) embedded radiative transfer





models into DL frameworks, enhancing the physical consistency, accuracy, and robustness of FSC estimation. These advancements underscore the growing potential of integrating DL with physically informed models to improve snow cover estimation in complex terrains. Overall, ML- and DL-based approaches have markedly advanced FSC retrieval by capturing high-dimensional nonlinear relationships among spectral, topographic, and auxiliary variables, thereby substantially enhancing

estimation accuracy and robustness.

Despite rapid progress in AI-driven FSC modelling, current studies still rely heavily on localized observations or limited experimental samples with narrow spatial coverage and short temporal spans. Although such models often achieve high accuracy in controlled settings, they struggle to scale effectively for large-area, long-term, and cross-regional FSC estimation. This limitation primarily stems from two interrelated challenges: **(1)** *Lack of standardized, interoperable AI-Ready FSC*

*datasets*. The absence of publicly available, large-scale, and temporally consistent FSC benchmark datasets severely constrains model training, generalization, and transferability across regions and time periods. **(2)** *Absence of unified standards for dataset construction and evaluation*. Existing FSC studies often adopt inconsistent protocols for reference label generation, feature selection, and data quality control, leading to heterogeneous data distributions and statistical biases. Moreover, the lack of universal evaluation metrics and standardized benchmarking frameworks undermines fair algorithm comparison and limits

the reproducibility and consistency of FSC products. Collectively, these issues highlight an urgent need to develop AI-ready FSC datasets through standardized, scalable, and interoperable workflows, thereby enabling intelligent, large-scale snow monitoring and improving methodological transparency and comparability across studies.

The concept of *AI-ready data* refers to high-quality, standardized, and interoperable datasets explicitly optimized for AI applications, i.e., data that are immediately usable for model training, validation, and deployment without extensive

preprocessing (Kidwai-Khan et al., 2024; Poduval et al., 2023). AI-ready datasets are characterized by *end-to-end curation*, encompassing data acquisition, cleaning, calibration, quality control, and metadata standardization to ensure traceability, reproducibility, and interoperability. The U.S. National Science Foundation's *National AI Research Resource (NAIRR) Pilot* exemplifies this paradigm, fostering open, cross-domain sharing of AI-ready data to advance education, research, and model development (NSF, 2024). These initiatives mark a broader global shift from traditional data archiving toward AI-oriented

data infrastructures designed to directly empower model-driven science. In the context of FSC mapping, constructing an AI-Ready FSC dataset entails systematically and intelligently organizing multi-source, heterogeneous snow-related environmental variables together with FSC reference labels into a cohesive, structured, and AI-oriented framework. Such a dataset enables direct AI model training without extensive preprocessing, facilitates reproducible and scalable FSC estimation, and supports transparent benchmarking and cross-regional generalization.

Building upon this concept, an AI-ready FSC dataset can be envisioned as a large-scale, standardized collection of snow cover samples that adhere to the following key principles: (1) **Spatiotemporal representativeness**. Samples should comprehensively capture diverse terrains, land-cover types, and snow conditions across multiple temporal scales. (2) **Physical and environmental completeness**. The dataset should incorporate well-defined environmental and geophysical variables that govern snow accumulation, melting, and redistribution processes. (3) **High-quality reference FSC labelling**. Reliable FSC





reference labels should be derived from multi-source, high-resolution remote sensing observations, with rigorous spatial and temporal consistency checks. (4) **Standardized metadata and AI-readiness evaluation protocols**. Comprehensive metadata and unified standards for dataset construction, documentation, and performance evaluation are essential to ensure reproducibility and comparability across studies. By adhering to these principles, an AI-Ready FSC dataset provides a robust foundation for advancing intelligent, physically consistent, and scalable FSC modelling, bridging the current gap between

algorithmic innovation and data standardization.

This study aims to develop the first standardized, AI-ready MODIS FSC dataset for China, termed **ChinaAI-FSC**, covering 22 snow seasons from 2000 to 2022. The dataset is constructed using high-resolution Landsat 5/7/8/9 and Sentinel-2 imagery as reference truth and integrates multi-source features, including MODIS surface reflectance (bands 1-7), topographic attributes, forest and land cover information, and geolocation factors. A total of 20 predictive features is derived and

systematically matched with reference FSC labels to form feature-FSC matchup samples. Each sample is organized into 128 × 128 MODIS-pixel image tiles, enabling both point-based and spatial (tile-level) modelling, as well as analyses of multi-scale spatial knowledge embedding. The ChinaAI-FSC dataset is designed to support systematic and fair evaluation of ML and DL models and to promote transparent, reproducible research through open access and standardized data formats. Building upon this dataset, the study explores three application scenarios: (1) assessing the potential of ChinaAI-FSC for AI-based FSC

estimation using six benchmark models (ANN, SVR, RF, CNN, UNet, and ResNet); (2) evaluating the accuracy and reliability of the standard MODIS FSC product using extensive reference samples from ChinaAI-FSC; and (3) conducting large-scale FSC mapping across China, particularly over complex terrain and spatially heterogeneous regions, to demonstrate the dataset's applicability to nationwide seamless FSC mapping. Overall, this study integrates multi-source remote sensing observations under a unified preprocessing and quality-control framework to produce standardized, high-consistency samples that serve as

a robust benchmark for AI model training and validation. The resulting dataset not only enhances cross-regional comparability and multi-model benchmarking but also provides a solid data foundation for studying cryosphere-climate-ecosystem interactions and for advancing operational snow monitoring, ultimately facilitating the generation of high-consistency, AI-driven FSC products for both scientific research and practical applications.

## 2 Study Area

This study focuses on the entire terrestrial domain of China, which exhibits remarkable representativeness and diversity in the global snow cover distribution. China's complex topography and diverse climate regimes give rise to three major stable snow regions: the Qinghai-Tibetan Plateau, the Northeast-Inner Mongolia, and Northern Xinjiang (Tan et al., 2019). The Qinghai-Tibetan Plateau is characterized by high elevations and alpine conditions, where snow dynamics are jointly influenced by monsoon and radiative processes. The Northeast-Inner Mongolia region, dominated by forested landscapes, displays strong

snow-vegetation interactions typical of mid- to high-latitude ecosystems. In contrast, Northern Xinjiang represents a continental mountain snow environment under the westerlies, exhibiting pronounced vertical stratification and high interannual

variability. Beyond these major snow regions, other parts of China, such as the Loess Plateau, the North China Plain, and the southwestern mountains, experience transient and intermittent snow events during winter. Overall, China encompasses highly diverse and contrasting snow regimes, ranging from persistent multi-year snow zones to short-lived seasonal snow cover. This

pronounced spatial and temporal heterogeneity provides an ideal foundation for constructing a comprehensive and representative AI-ready FSC dataset capable of supporting large-scale and multi-scenario modelling. Accordingly, the study domain is divided into four subregions: Xinjiang (XJ), Qinghai-Tibetan Plateau (QTP), Northeast-Inner Mongolia (NE-IM), and Other regions (Fig. 1).

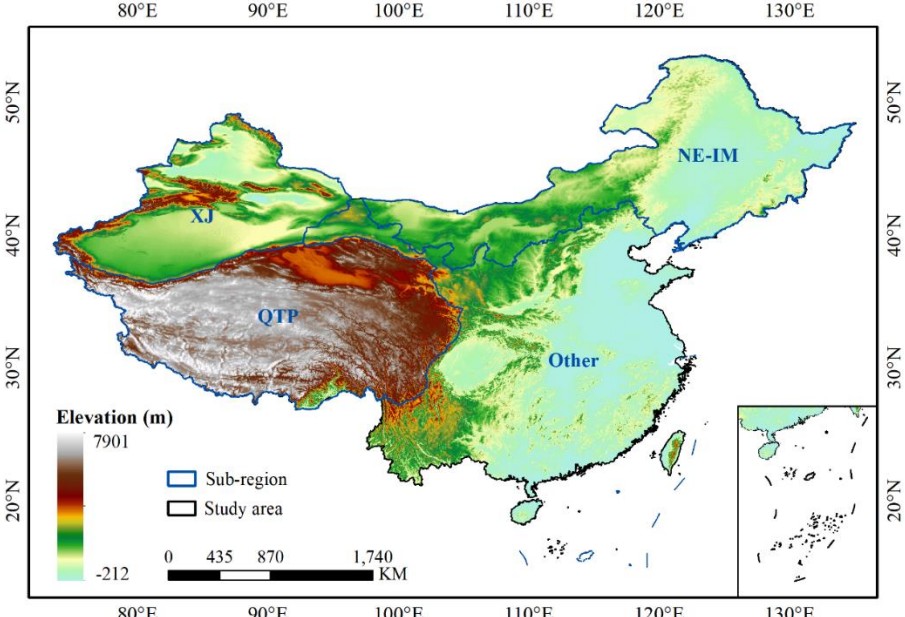

**Figure 1: Overview of the study area. The entire study domain is divided into four subregions: Xinjiang (XJ), Qinghai-Tibetan Plateau (QTP), Northeast-Inner Mongolia (NE-IM), and Other regions.**

## 3 Development of the AI-Ready MODIS FSC Dataset

This study aims to construct the first standardized, AI-ready MODIS FSC sample database for China (ChinaAI-FSC), spanning 22 snow seasons (from October to March of the following year) from 2000 to 2022. The detailed construction workflow is

illustrated in Fig. 2.



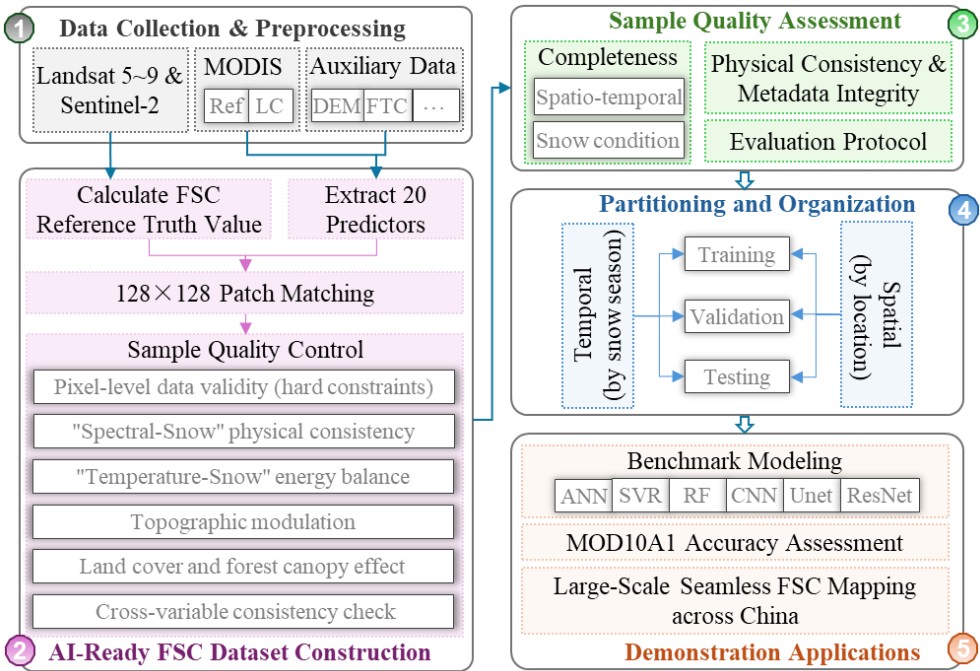

Figure 2: Workflow for constructing the AI-ready MODIS FSC sample dataset.

## 3.1 Data Collection & Preprocessing

### 3.1.1 Landsat and Sentinel-2 imagery

The Landsat imagery used in this study, including Landsat-5 TM, Landsat-7 ETM+, Landsat-8 OLI, and Landsat-9 OLI-2, was obtained from the U.S. Geological Survey (USGS) Earth Explorer platform (https://earthexplorer.usgs.gov/). Sentinel-2 MSI imagery was sourced from the European Space Agency (ESA) Copernicus Open Access Hub (https://scihub.copernicus.eu/). To ensure data quality and inter-sensor consistency, only clear-sky scenes with a cloud cover fraction below 15% were selected across 22 snow seasons between 2000 and 2022. The number of raw scenes collected for

each subregion and snow season is summarized in Table 1.

   To ensure comparability and consistency of multi-source remote sensing imagery, systematic preprocessing and cross-sensor harmonization were performed for the Landsat and Sentinel-2 datasets. First, all scenes were radiometrically calibrated and atmospherically corrected to convert the original digital numbers (DNs) into surface reflectance, thereby minimizing atmospheric scattering and absorption effects. Topographic correction was subsequently applied using the C-correction method

to mitigate terrain-induced illumination variations (Crawford et al., 2023). For cloud and shadow detection, the CFMask algorithm was employed for Landsat imagery to generate reliable cloud and shadow masks (Crawford et al., 2023), whereas Sentinel-2 imagery was processed using the Scene Classification Layer (SCL) in conjunction with the MAJA algorithm to achieve accurate cloud and shadow identification and removal (Crawford et al., 2023; Main-Knorn et al., 2017). For



Landsat-7 ETM+ imagery affected by the Scan Line Corrector (SLC) failure after May 2003, missing scan lines were spatially
interpolated using a local neighbourhood approach to restore spatial continuity and completeness (Markham et al., 2004).
Additionally, due to orbit drift and the consequent degradation of observation quality after 2021 (Qiu et al., 2021), Landsat-7
ETM+ data acquired from 2021 onwards were excluded. Next, to address radiometric discrepancies among sensors arising
from differences in spectral response and calibration characteristics, cross-sensor normalization was performed using Landsat-
9 OLI-2 as the radiometric reference. Overlapping image pairs were leveraged along with regression-based adjustments to
harmonize Landsat-8/7/5 and Sentinel-2 MSI surface reflectance, ensuring radiometric consistency and temporal comparability
across the entire time series (Chander et al., 2013; Claverie et al., 2018). Finally, multi-sensor mosaicking was conducted for
images acquired on the same day. In cases of spatial overlap, pixels from the most recent sensor with superior radiometric
calibration were preferentially retained. The mosaicked imagery was then reprojected to a geographic coordinate system with
a uniform spatial resolution of 0.00833° (~30 m) to ensure spatial consistency across the dataset.

**Table 1: Number of Landsat and Sentinel-2 Images Used for 2000-2022 Snow Seasons**

| Snow Season | XJ | | | | | NE-IM | | | | |
|---|---|---|---|---|---|---|---|---|---|---|
| | LT05 | LE07 | LC08 | LC09 | S2 | LT05 | LE07 | LC08 | LC09 | S2 |
| 2000-2001 | 127 | 265 | - | - | - | 342 | 565 | - | - | - |
| 2001-2002 | 169 | 222 | - | - | - | 446 | 436 | - | - | - |
| 2002-2003 | 143 | 280 | - | - | - | 474 | 648 | - | - | - |
| 2003-2004 | 107 | 215 | - | - | - | 414 | 131 | - | - | - |
| 2004-2005 | 142 | 235 | - | - | - | 509 | 64 | - | - | - |
| 2005-2006 | 74 | 273 | - | - | - | 264 | 62 | - | - | - |
| 2006-2007 | 143 | 249 | - | - | - | 569 | 83 | - | - | - |
| 2007-2008 | 19 | 280 | - | - | - | 61 | 305 | - | - | - |
| 2008-2009 | 184 | 306 | - | - | - | 202 | 175 | - | - | - |
| 2009-2010 | 275 | 274 | - | - | - | 190 | 260 | - | - | - |
| 2010-2011 | 331 | 313 | - | - | - | 260 | 462 | - | - | - |
| 2011-2012 | 29 | 376 | - | - | - | 7 | 452 | - | - | - |
| 2012-2013 | - | 306 | - | - | - | - | 350 | - | - | - |
| 2013-2014 | - | 325 | 367 | - | - | - | 438 | 474 | - | - |
| 2014-2015 | - | 353 | 408 | - | - | - | 483 | 522 | - | - |
| 2015-2016 | - | 427 | 316 | - | - | - | 501 | 629 | - | - |
| 2016-2017 | - | 318 | 272 | - | - | - | 538 | 567 | - | - |
| 2017-2018 | - | 385 | 405 | - | 42 | - | 476 | 393 | - | 63 |
| 2018-2019 | - | 380 | 433 | - | 1160 | - | 276 | 369 | - | 1219 |
| 2019-2020 | - | 409 | 416 | - | 2089 | - | 329 | 586 | - | 3749 |
| 2020-2021 | - | 184 | 299 | - | 1951 | - | 121 | 647 | - | 2558 |
| **2021-2022** | - | - | 421 | 189 | 2595 | - | - | 590 | 419 | 1914 |
| **Total** | 1743 | 6375 | 3337 | 189 | 7837 | 3738 | 7155 | 4777 | 419 | 9503 |

| Snow Season | QTP | | | | | Other | | | | | China |
|---|---|---|---|---|---|---|---|---|---|---|---|
| | LT05 | LE07 | LC08 | LC09 | S2 | LT05 | LE07 | LC08 | LC09 | S2 | |
| 2000-2001 | 466 | 491 | - | - | - | 146 | 109 | - | - | - | 2511 |
| 2001-2002 | 424 | 488 | - | - | - | 143 | 103 | - | - | - | 2431 |
| 2002-2003 | 324 | 561 | - | - | - | 158 | 204 | - | - | - | 2792 |
| 2003-2004 | 396 | 345 | - | - | - | 131 | 201 | - | - | - | 1940 |
| 2004-2005 | 449 | 340 | - | - | - | 174 | 193 | - | - | - | 2106 |
| 2005-2006 | 227 | 574 | - | - | - | 70 | 175 | - | - | - | 1719 |
| 2006-2007 | 483 | 376 | - | - | - | 120 | 170 | - | - | - | 2193 |





| | LT05 | LE07 | LC08 | LC09 | S2 | LT05 | LE07 | LC08 | LC09 | S2 | Total |
|---|---|---|---|---|---|---|---|---|---|---|---|
| 2007-2008 | 155 | 468 | - | - | - | 35 | 175 | - | - | - | 1498 |
| 2008-2009 | 606 | 630 | - | - | - | 90 | 121 | - | - | - | 2314 |
| 2009-2010 | 539 | 576 | - | - | - | 113 | 138 | - | - | - | 2365 |
| 2010-2011 | 606 | 564 | - | - | - | 119 | 113 | - | - | - | 2768 |
| 2011-2012 | 97 | 655 | - | - | - | 15 | 116 | - | - | - | 1747 |
| 2012-2013 | - | 695 | - | - | - | - | 183 | - | - | - | 1534 |
| 2013-2014 | - | 721 | 767 | - | - | - | 160 | 98 | - | - | 3350 |
| 2014-2015 | - | 677 | 540 | - | - | - | 103 | 86 | - | - | 3172 |
| 2015-2016 | - | 676 | 274 | - | - | - | 105 | 123 | - | - | 3051 |
| 2016-2017 | - | 645 | 702 | - | - | - | 101 | 95 | - | - | 3238 |
| 2017-2018 | - | 237 | 747 | - | 166 | - | 141 | 131 | - | 25 | 3211 |
| 2018-2019 | - | 310 | 698 | - | 628 | - | 113 | 102 | - | 174 | 5862 |
| 2019-2020 | - | 340 | 271 | - | 1320 | - | 182 | 122 | - | 467 | 10280 |
| 2020-2021 | - | 219 | 521 | - | 1255 | - | 88 | 134 | - | 400 | 8377 |
| 2021-2022 | - | - | 670 | 338 | 4201 | - | - | 102 | 63 | 542 | 12044 |
| **Total** | **4772** | **10588** | **5190** | **338** | **7570** | **1314** | **2994** | **993** | **63** | **1608** | **80503** |

Note: LT05, LE07, LC08, LC09, and S2 denote Landsat-5 TM, Landsat-7 ETM+, Landsat-8 OLI, Landsat-9 OLI-2, and Sentinel-2, respectively

### 3.1.2 MODIS data

The MODIS data products utilized in this study include surface reflectance bands 1-7, the standard MODIS snow product (MOD10A1, Collection 6), and the MODIS land cover dataset (MCD12Q1, Collection 6). The surface reflectance data was obtained from the Global 500 m seamless dataset of MODIS-derived land surface reflectance (SDC500) for 2000-2022, produced by Liang et al. (2024), which demonstrates a mean absolute error (MAE) of only 0.043. All MODIS datasets were subsequently reprojected and resampled to a common geographic coordinate system with a spatial resolution of 0.005°
(~500 m).

### 3.1.3 Auxiliary Data

Additional environmental factors influencing snow distribution were incorporated, including forest cover, land surface temperature (LST), and topographic variables derived from digital elevation data (DEM). Forest cover was represented using the global annual fractional tree cover dataset for 2000-2021 at 250 m resolution (GLOBMAP FTC), which accurately captures
both global and regional forest dynamics (Liu et al., 2024). LST data were obtained from a daily 1 km all-weather land surface temperature dataset over China and surrounding regions (TRIMS LST), which shows high agreement with MODIS LST products in both magnitude and spatial distribution, with mean daytime and nighttime biases of 0.09 K and -0.03 K, and standard deviations of 1.45 K and 1.17 K, respectively (Tang et al., 2024). Topographic information was derived from the Shuttle Radar Topography Mission (SRTM) Version 4.1 DEM, accessed via the CGIAR-CSI database (Jarvis et al., 2008).
Considering the pronounced seasonal periodicity of snow cover, Julian day was also included as an auxiliary variable. All auxiliary datasets were resampled to match the spatial resolution and coordinate system of the MODIS land surface products.



## 3.2 AI-Ready FSC Dataset Construction

### 3.2.1 Calculation of Reference FSC

The standard SNOMAP algorithm (Hall et al., 1995) derives binary snow cover from the Normalized Difference Snow Index
(NDSI), calculated using visible and shortwave infrared reflectance. However, in forested regions, snow detection accuracy
can be substantially reduced by canopy occlusion and mixed-pixel effects, resulting in underestimation and omission errors.
To address these limitations, Klein et al. (1998) proposed an improved SNOMAP algorithm that integrates a canopy reflectance
model into the original NDSI framework. This modification explicitly separates the reflectance contributions of the canopy
and the underlying snow surface through canopy reflectance and transmittance, thereby improving snow detection accuracy in
vegetated areas. In this study, the improved SNOMAP algorithm was applied to the pre-processed Landsat and Sentinel-2
imagery to generate binary snow maps at 30 m spatial resolution. Based on these high-resolution snow maps, FSC reference
values at the MODIS scale were estimated. For each MODIS pixel, a circular neighbourhood was centred on the MODIS pixel
centroid with a radius equal to 1.5 times the MODIS pixel size, to account for potential geolocation discrepancies between
MODIS and Landsat/Sentinel-2 imagery. The FSC reference value was then computed as the proportion of snow-covered
Landsat/Sentinel-2 pixels within this neighbourhood (Dobreva and Klein, 2011).

### 3.2.2 Extraction of Feature Variables

A total of twenty feature variables were derived for FSC modelling (Table 2), including MODIS surface reflectance bands 1-
7 (denoted as Ref1-Ref7) from the SDC500 product, NDSI, Normalized Difference Vegetation Index (NDVI), land cover (LC),
LST, FTC, and topographic factors. NDSI and NDVI were computed from the SDC500 reflectance using bands 4 and 6, and
bands 2 and 1, respectively. Five topographic variables, i.e., elevation, slope, aspect, terrain relief, and surface roughness, were
derived from the DEM.

**Table 2: Description of the 20 Input Features Integrated in the AI-Ready FSC Dataset**

| Variable | Data Product | Data Source | Reference |
|---|---|---|---|
| Ref1-Ref7 | SDC500 | https://data-starcloud.pcl.ac.cn/iearthdata/27 | Liang et al. (2024) |
| NDSI | | | - |
| NDVI | | | - |
| LC | MCD12Q1 | https://lpdaac.usgs.gov/products/mcd12q1v061/ | - |
| LST | TRIMS LST | https://data.tpdc.ac.cn/zh-hans/data/05d6e569-6d4b-43c0-96aa-5584484259f0 | Tang et al. (2024) |
| FTC | GLOBMAP FTC | https://zenodo.org/records/10589730 | Liu et al. (2024) |
| Elevatio, Slope, Aspect, Terrain Relief, Surface Roughness | SRTM DEM | https://srtm.csi.cgiar.org/srtmdata/ | Jarvis et al. (2008) |
| Longitude, Latitude | - | - | - |
| Julian Day | - | - | - |



### 3.2.3 Generation of the Original FSC Sample Dataset

Considering the distinct structural requirements of different ML and DL models, the sample was organized to accommodate
both point-based and spatially continuous inputs. Traditional models such as ANNs and SVR typically operate on discrete
point samples, whereas convolution-based models (e.g., CNNs) require spatially continuous image blocks. Considering the
spatial resolution of MODIS data and the spatial coverage of individual high-resolution Landsat 5/7/8/9 and Sentinel-2 scenes,
the study area was divided into regular 0.64° × 0.64° grid tiles, each corresponding to 128 × 128 MODIS pixels. Each tile was
assigned a unique row-column identifier based on its geographic position (Fig. 3). This spatial partitioning provides an optimal
balance between areal coverage and computational efficiency while maintaining high decomposability. It also supports flexible
aggregation into multi-scale tiles (e.g., 8×8, 16×16, 32×32, and 64×64 MODIS pixels), enabling adaptive feature extraction
and model training across different spatial scales. Within each grid tile, twenty feature variables (input features) were extracted
from MODIS and auxiliary datasets and paired with FSC reference values derived from Landsat/Sentinel-2 observations based
on precise spatiotemporal correspondence. These "feature-reference FSC" matchups collectively form the original MODIS
FSC sample dataset. Over the 22 snow seasons, a total of **166,763 original samples** (each comprising 128×128 MODIS pixels)
were generated (Table 3).

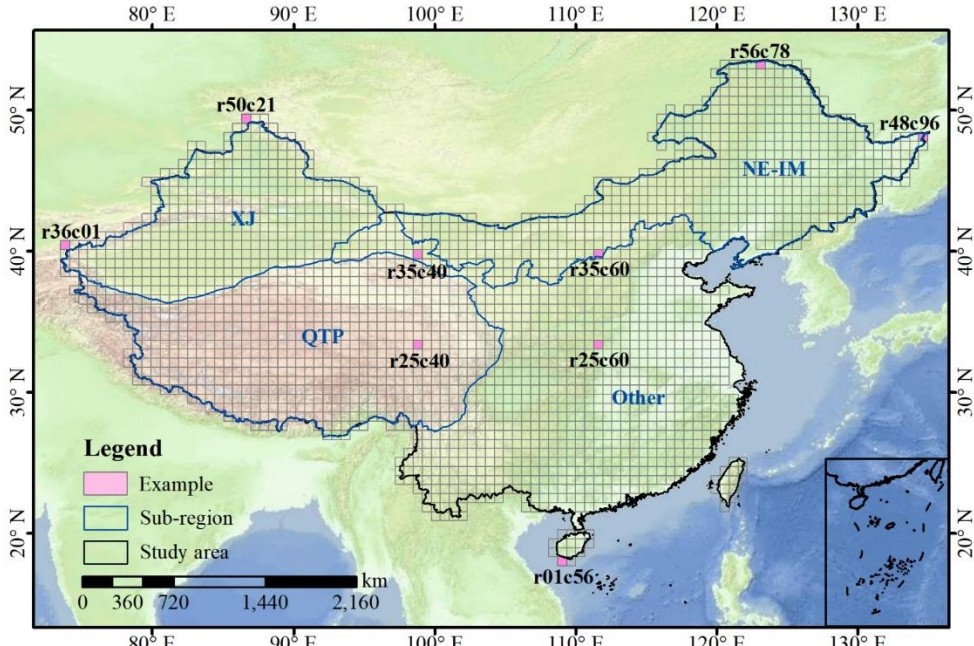

**Figure 3: Spatial partitioning of the study area into regular 0.64° × 0.64° grid tiles, each corresponding to 128 × 128 MODIS pixels.
Tiles are assigned unique row and column identifiers based on spatial location, with column numbers (c01, c02, …) increasing from**
**west to east and row numbers (r01, r02, …) increasing from south to north.**

**Table 3: Statistics of Original FSC Samples over 22 Snow Seasons**

| Snow Season | XJ | NE-IM | QTP | Other | China |
|---|---|---|---|---|---|
| 2000-2001 | 1107 | 3142 | 3038 | 395 | 7682 |



| | | | | |
|---|---|---|---|---|
| 2001-2002 | 1047 | 2892 | 2899 | 343 | 7181 |
| 2002-2003 | 1209 | 3477 | 2856 | 652 | 8194 |
| 2003-2004 | 1076 | 1608 | 2319 | 440 | 5443 |
| 2004-2005 | 1208 | 1846 | 2516 | 609 | 6179 |
| 2005-2006 | 1146 | 968 | 2498 | 344 | 4956 |
| 2006-2007 | 1269 | 2172 | 2725 | 424 | 6590 |
| 2007-2008 | 1064 | 1112 | 1992 | 398 | 4566 |
| 2008-2009 | 1324 | 1136 | 3912 | 256 | 6628 |
| 2009-2010 | 1375 | 1453 | 3612 | 394 | 6834 |
| 2010-2011 | 1741 | 2417 | 3730 | 302 | 8190 |
| 2011-2012 | 1162 | 1457 | 2380 | 202 | 5201 |
| 2012-2013 | 889 | 1117 | 2260 | 296 | 4562 |
| 2013-2014 | 1618 | 2975 | 4688 | 388 | 9669 |
| 2014-2015 | 1817 | 3185 | 3690 | 223 | 8915 |
| 2015-2016 | 1737 | 3298 | 2836 | 356 | 8227 |
| 2016-2017 | 1389 | 3568 | 4053 | 237 | 9247 |
| 2017-2018 | 1807 | 2717 | 2848 | 441 | 7813 |
| 2018-2019 | 2518 | 2286 | 2853 | 327 | 7984 |
| 2019-2020 | 3006 | 5339 | 1966 | 653 | 10964 |
| 2020-2021 | 2504 | 4222 | 2429 | 485 | 9640 |
| 2021-2022 | 3257 | 4535 | 3726 | 580 | 12098 |
| **Total** | **35270** | **56922** | **65826** | **8745** | **166763** |

### 3.2.4 Sample Quality Control

To guarantee the physical consistency and high reliability of the FSC sample dataset, a comprehensive quality control (QC) framework was established by integrating domain-specific physical constraints with multi-level consistency checks. The QC process was designed at two hierarchical levels, i.e., pixel and tile, to form a transparent and reproducible quality assurance chain. At the pixel level, physical plausibility screening was performed to eliminate invalid or inconsistent observations. At the tile level, multiple constraint-based checks were applied, including spectral and energy balance consistency to ensure snow-related radiative coherence, topography- and LC-based modulation to mitigate spatial heterogeneity, and cross-variable consistency validation to detect and remove physically implausible samples. These procedures collectively ensured that the resulting dataset is physically coherent, statistically robust, and fully AI-ready for subsequent model training and evaluation (Fig. 4).





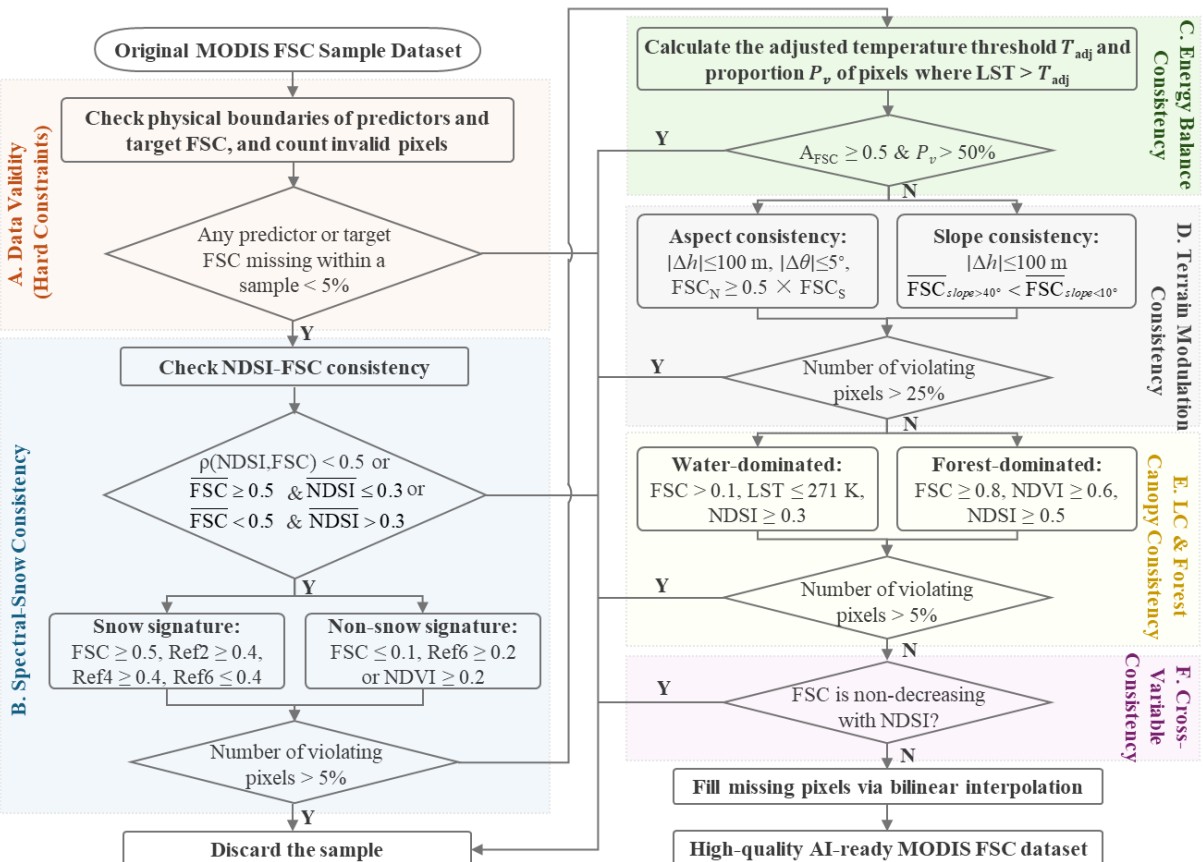

**Figure 4: Technical Workflow of FSC Sample Quality Control.**

**A. Pixel-level data validity (hard constraints)**

Pixel-level validity checks were conducted to eliminate physically inconsistent or anomalous samples. Specifically, physical range constraints were imposed on all feature variables and the target FSC variable. For instance, surface reflectance for all bands was required to satisfy $0 \leq \rho \leq 1$; NDVI $\in$ [-1, 1]; NDSI $\in$ [-1, 1]; LST $\in$ [180 K, 340 K]; FTC $\in$ [0, 1]; FSC $\in$ [0, 1]; and slope $\in$ [0°, 60°] (slopes > 60° typically indicate noise or projection errors). Pixels exceeding these limits were flagged as invalid. Additionally, for each sample tile, if the proportion of missing pixels in any variable exceeded 5%, the sample was

discarded to ensure data validity and statistical reliability.

**B. Spectral-Snow Physical Consistency**

To ensure the consistency between spectral characteristics and snow physical properties, two complementary validation steps were implemented: NDSI-FSC relationship verification and spectral signature constraint checking.

①  **NDSI-FSC Consistency:** At the pixel level, if the correlation coefficient between NDSI and the reference FSC values,

$\rho$(NDSI, FSC), within a sample was less than 0.5, the sample was considered spectrally inconsistent with snow physics. At the



tile level, if the tile-averaged FSC, i.e., $\overline{\mathrm{FSC}} \geq 0.5$, but the tile-averaged NDSI, i.e., $\overline{\mathrm{NDSI}} \leq 0.3$, or if the $\overline{\mathrm{FSC}} < 0.5$ while the $\overline{\mathrm{NDSI}} > 0.3$, the sample was also regarded as inconsistent and was removed.

② **Spectral Signatures of Snow Pixels:** For snow pixels (FSC $\geq$ 0.5), spectral reflectance was required to satisfy the following physical constraints: green band reflectance Ref4 $\geq$ 0.2, near-infrared reflectance Ref2 $\geq$ 0.4, and shortwave infrared reflectance Ref6 $\leq$ 0.4. If more than 5% of pixels in a sample violated any of these conditions, the sample was excluded.

③ **Spectral Signatures of Non-Snow Pixels:** For non-snow pixels (FSC $\leq$ 0.1), the conditions Ref6 $\geq$ 0.2 or NDVI $\geq$ 0.2 must both be met. Samples were discarded if more than 5% of pixels failed to meet these constraints.

**C. Temperature-Snow Energy Balance Consistency**

To assess the consistency of energy balance, the relationship between LST and snow cover was evaluated. This study applied an elevation-corrected cooling constraint by calculating a pixel-wise adjusted temperature threshold $T_{adj}$ ( Eq. (1)) using an elevation aligned with the LST data. Specifically, it was assumed that a 1,000 m increase in elevation corresponds to a 2 K increase in the threshold, making the consistency criterion more lenient at higher elevations. The proportion of pixels within each sample exceeding this threshold, denoted as $P_v$, was then computed. Samples with a mean FSC $\geq$ 0.5 and $P_v > 0.5$ were considered inconsistent with the energy balance after accounting for elevation effects and were consequently removed from the dataset.

$$T_{adj}(i,j) = T_{base} + 2K \times \frac{h(i,j)}{1000m} \tag{1}$$

Here, $T_{base}$ is the base temperature threshold, set to 273.15 K, and $h(i,j)$ denotes the elevation of pixel $(i,j)$.

**D. Topographic Modulation Consistency**

Considering the significant influence of topographic factors (slope and aspect) on snow distribution, the sample screening is conducted as follows:

① **Aspect Consistency**: Under conditions of similar elevation (±100 m) and slope (±5°), the FSC on north-facing (315°-45°) shaded slopes should be no less than 50% of that on south-facing (135°-225°) sun-exposed slopes. If more than 25% of the pixels within a sample violate this criterion, the sample is regarded as aspect-inconsistent and is excluded (Eq. (2)).

$$\begin{cases} |\Delta h| \leq 100m \\ |\Delta \theta| \leq 5° \\ FSC_{north} \geq 0.5 \times FSC_{south} \end{cases}$$

② **Slope Consistency**: Under comparable elevation conditions, the average FSC on steep slopes (>40°) should be lower than that on gentle slopes (<10°). If more than 25% of the pixels within a sample violate this criterion, the sample is considered slope-inconsistent and is excluded (Eq. (3)).

$$\begin{cases} |\Delta h| \leq 100m \\ \overline{FSC}_{slope>40°} < \overline{FSC}_{slope<10°} \end{cases} \tag{3}$$





**E. Land Cover and Forest Canopy Effect Consistency**

①**Water Body Consistency Constraint:** Samples with >60% water pixels are classified as water-dominated. For such samples, any pixels with FSC > 0.1 must satisfy the physical conditions of LST ≤ 271 K and NDSI ≥ 0.3 to represent possible freezing or ice formation. If more than 5% of pixels violate this criterion, the sample is considered spectrally and thermally inconsistent and is excluded.

②**Forest Canopy Obstruction Constraint:**

Samples with FTC ≥ 60% are classified as forest-dominated, where snow signals are prone to systematic underestimation due to canopy obstruction. For these samples, if the proportion of pixels with high snow cover (FSC ≥ 0.8) exceeds 5% and these pixels simultaneously exhibit NDVI ≥ 0.6 and NDSI ≥ 0.5 (i.e., indicating a physically implausible combination of high snow cover under dense vegetation), the sample is regarded as inconsistent with canopy-obstructed snow detection and is excluded.

**F. Cross-Variable Consistency Check**

The consistency between FSC and NDSI is evaluated. Pixels within each sample are first divided into 10 equal-width bins based on NDSI values, and the average FSC is calculated for each bin. In theory, the mean FSC should increase monotonically with NDSI. If two or more clear reversals occur across the NDSI range (i.e., the average FSC decreases by more than 0.1), the sample is considered anomalous and is excluded to ensure data consistency.

Table 4 summarizes how the number of samples changed during the six quality control steps. After applying these six rigorous

QC steps described above, the few remaining missing pixels in the samples were filled using bilinear interpolation. The resulting AI-ready, high-quality MODIS FSC sample dataset is summarized in Table 5.

**Table 4: Number of original, excluded, and remaining samples after each QC step.**

| QC steps | XJ | | NE-IM | | QTP | | Other | | China | |
|---|---|---|---|---|---|---|---|---|---|---|
| | E | R | E | R | E | R | E | R | E | R |
| **Original** | | 35270 | | 56922 | | 65826 | | 8745 | | 166763 |
| **A** | 20454 | 14816 | 31462 | 25460 | 30394 | 35432 | 8010 | 735 | 90320 | 76443 |
| **B.①** | 1446 | 13370 | 4484 | 20976 | 2934 | 32498 | 69 | 666 | 8933 | 67510 |
| **B.②** | 1669 | 11701 | 2304 | 18672 | 354 | 32144 | 0 | 666 | 4327 | 63183 |
| **B.③** | 2692 | 9009 | 1525 | 17147 | 8220 | 23924 | 308 | 358 | 12745 | 50438 |
| **C** | 872 | 8137 | 9 | 17138 | 0 | 23924 | 0 | 358 | 881 | 49557 |
| **D.①** | 484 | 7653 | 208 | 16830 | 518 | 23406 | 1 | 357 | 1211 | 48246 |
| **D.②** | 0 | 7653 | 0 | 16830 | 0 | 23406 | 0 | 357 | 0 | 48246 |
| **E.①** | 0 | 7653 | 0 | 16830 | 0 | 23406 | 0 | 357 | 0 | 48246 |
| **E.②** | 0 | 7653 | 0 | 16830 | 0 | 23406 | 0 | 357 | 0 | 48246 |
| **F** | 93 | 7560 | 299 | 16531 | 124 | 23282 | 2 | 355 | 518 | 47728 |

Note: E and R indicate, respectively, the number of samples excluded and the number of samples remaining at each step.

**Table 5: Statistics of AI-Ready MODIS FSC samples after quality control.**

| Snow Season | XJ | NE-IM | QTP | Other | China |
|---|---|---|---|---|---|
| 2000-2001 | 267 | 1081 | 1116 | 9 | 2473 |
| 2001-2002 | 232 | 793 | 1007 | 9 | 2041 |
| 2002-2003 | 284 | 1012 | 1231 | 48 | 2575 |
| 2003-2004 | 165 | 436 | 786 | 22 | 1409 |
| 2004-2005 | 177 | 609 | 930 | 25 | 1741 |
| 2005-2006 | 197 | 325 | 837 | 7 | 1366 |
| 2006-2007 | 234 | 640 | 1025 | 8 | 1907 |
| 2007-2008 | 219 | 352 | 751 | 19 | 1341 |



| | | | | | |
|---|---|---|---|---|---|
| 2008-2009 | 313 | 326 | 1601 | 4 | 2244 |
| 2009-2010 | 290 | 313 | 1376 | 18 | 1997 |
| 2010-2011 | 387 | 656 | 1287 | 7 | 2337 |
| 2011-2012 | 250 | 423 | 836 | 9 | 1518 |
| 2012-2013 | 181 | 355 | 739 | 33 | 1308 |
| 2013-2014 | 314 | 1035 | 1660 | 8 | 3017 |
| 2014-2015 | 410 | 930 | 1256 | 1 | 2597 |
| 2015-2016 | 384 | 952 | 865 | 26 | 2227 |
| 2016-2017 | 229 | 1113 | 1195 | 7 | 2544 |
| 2017-2018 | 360 | 737 | 813 | 0 | 1910 |
| 2018-2019 | 466 | 612 | 1070 | 8 | 2156 |
| 2019-2020 | 747 | 1411 | 824 | 30 | 3012 |
| 2020-2021 | 595 | 1062 | 640 | 25 | 2322 |
| 2021-2022 | 859 | 1358 | 1437 | 32 | 3686 |
| **Total** | **7560** | **16531** | **23282** | **355** | **47728** |

### 3.3 Sample Quality Assessment

To guarantee the robustness and AI-readiness of the constructed MODIS FSC sample database, this study developed a multi-dimensional sample quality assessment framework. Beyond traditional completeness and consistency checks, the framework introduces a novel set of AI-Readiness evaluation protocols that systematically quantify the degree to which a dataset supports AI-driven modelling. These protocols not only serve as an objective metric for assessing AI-readiness and provide a unified and standardized foundation for evaluating the quality, usability, and scientific reliability of the ChinaAI-FSC dataset, but also have potential applicability to other Earth observation datasets aiming for AI-ready standards.

### 3.3.1 Sample Completeness

The completeness of the AI-ready FSC dataset was evaluated in terms of spatial, temporal, and snow-cover representativeness to ensure the absence of systematic bias or tilt. Spatial completeness was assessed by examining whether the sample distribution adequately represents the entire spatial extent of China, with particular emphasis on achieving full coverage of the three major stable snow regions. Fig. 5a-b compares the spatial distributions of samples before and after QC. Although QC inevitably reduced the total number of samples, the retained dataset preserved broad and balanced coverage across the three principal snow regions. Consequently, the post-QC samples maintain sufficient spatial representativeness for model training and regional-scale analyses

Temporal completeness was evaluated by analysing the temporal distribution of samples to verify consistent representation across different snow seasons and years. As summarized in Table 5, all four subregions within the study area exhibit relatively balanced sample distributions across the 22 snow seasons, ensuring that the dataset adequately represents temporal variations in snow-cover conditions. This temporal balance ensures that the AI-ready MODIS FSC dataset provides a stable and unbiased foundation for long-term snow monitoring and model development.

Snow-cover completeness was assessed with respect to the full FSC range (0-100%), ensuring adequate representation of low-, medium-, and high-fraction snow conditions. As illustrated in Fig. 5c-d, the original dataset displayed a highly imbalanced FSC distribution, with most samples concentrated in low-FSC intervals (0-10%), particularly across the QTP, while high-FSC

samples (>80%) were relatively sparse. This pattern reflects the inherent spatiotemporal heterogeneity of snow occurrence across China's diverse climatic zones. Following QC, although the total number of samples decreased substantially, the FSC
distribution became markedly more balanced, with extreme bins significantly reduced and intermediate categories better represented.

To further evaluate whether the post-QC dataset retained the statistical structure of the original data, kernel density analyses were conducted across all 20 feature variables (Fig. 6). The resulting density curves show strong consistency between the quality-controlled and original datasets in terms of overall shape, central tendency, skewness, and tail behaviour. This close
correspondence demonstrates that the QC procedures successfully preserved the statistical properties and structural integrity of the original dataset without introducing sampling bias or systematic deviation. The comparable density profiles confirm that the final AI-ready FSC dataset achieves enhanced regional representativeness, balanced FSC category coverage, and reliable statistical continuity, providing a robust foundation for subsequent modelling and large-scale snow-cover estimation.

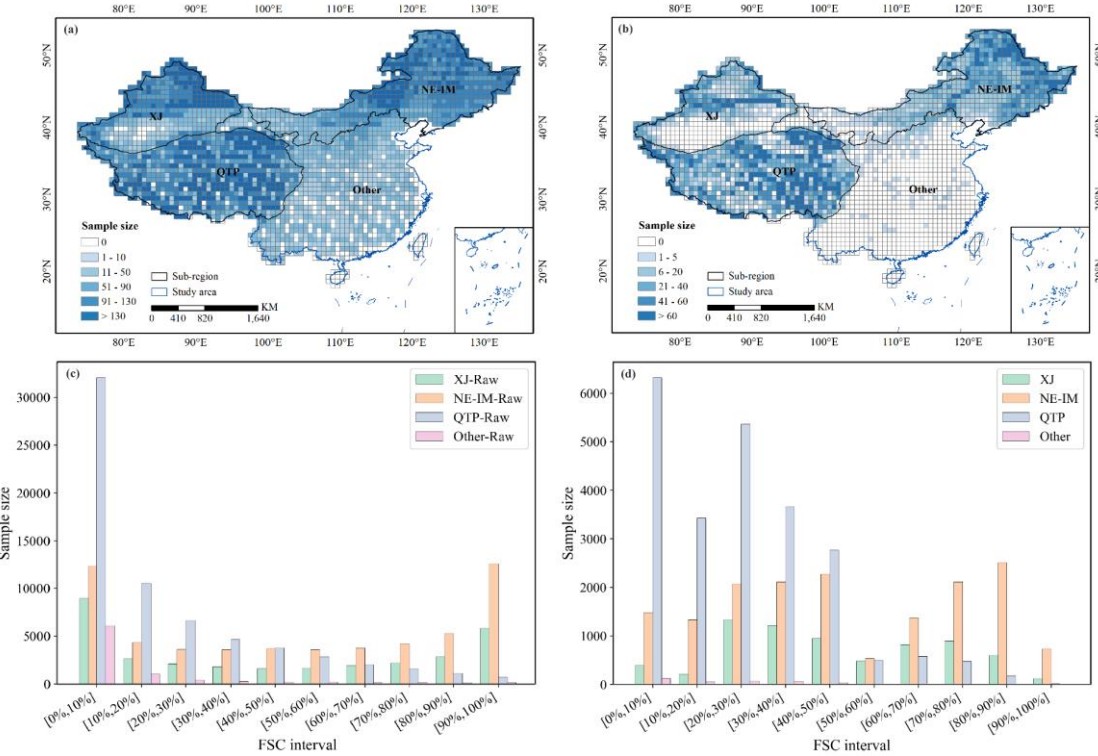

**Figure 5: Spatial and snow condition completeness of the MODIS FSC sample dataset. a) and (b) show the spatial distribution of the original and the post-QC AI-ready FSC dataset, respectively. (c) and (d) illustrate the distributions of FSC values before and after quality control, respectively.**

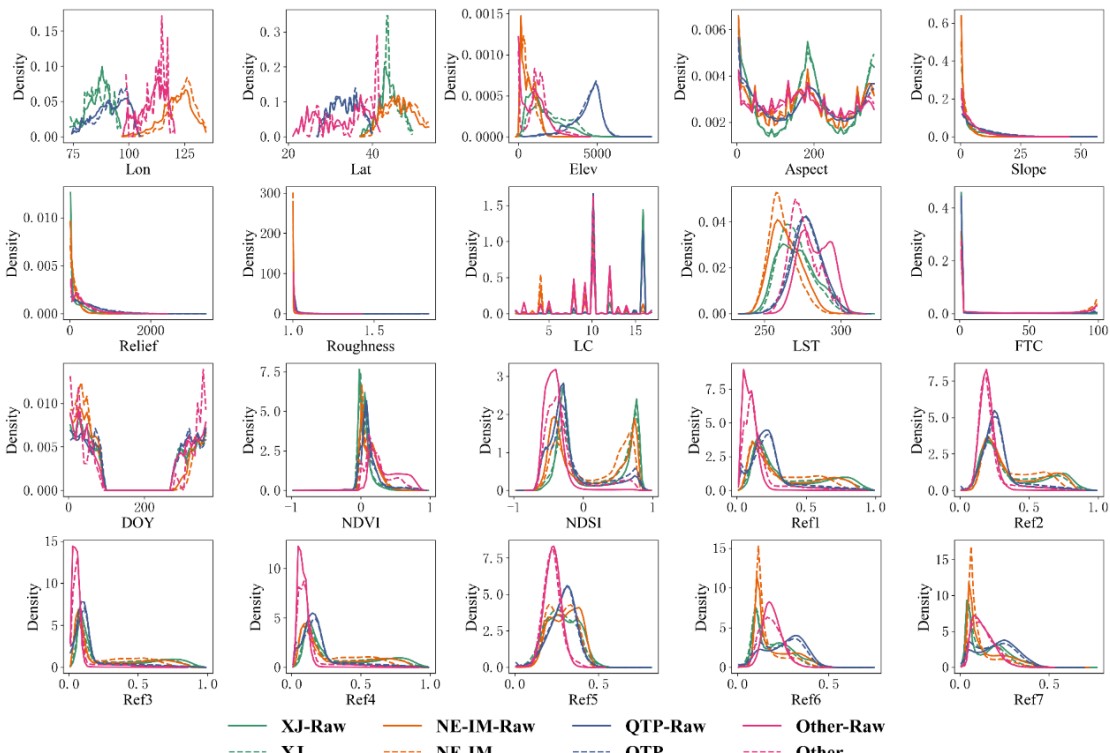

**Figure 6: Kernel density distributions of the original and post-QC MODIS FSC samples**

### 3.3.2 Metadata Integrity and Physical Consistency

The quality of the AI-Ready FSC dataset was further assessed in terms of metadata integrity and physical consistency, emphasizing standardized documentation, traceability, and the physical plausibility of feature-target relationships. The AI-Ready MODIS FSC dataset is distributed in GeoTIFF format, comprising single-band FSC reference files and multi-band feature files. It covers mainland China (70°E-140°E, 15°N-55°N) over 22 snow seasons (2000-2022) at a spatial resolution of 0.005°, referenced to the WGS84 geographic coordinate system. Each feature file consists of 128×128 pixels with 20 feature bands representing geographical, topographical, and remote-sensing variables, including longitude, latitude, elevation, aspect, slope, terrain relief, surface roughness, LC, LST, FTC, DOY, NDVI, NDSI, and seven reflectance bands (Ref1-Ref7), respectively. Each sample strictly adheres to **unified naming conventions** describing its spatial location, acquisition time, snow-cover extent, and associated feature variables. For example, files named "20201020_r37c06_4.tif" and "20201020_r37c06_4_FSC.tif" correspond to the feature and reference FSC files, respectively, for the sample located at row 37, column 6 on 20 October 2020 (see Fig. 3 for the spatial indexing scheme). The numeric suffix denotes the mean FSC percentile range of the sample, with mean FSC values discretized into ten intervals: [0,10%], (10%,20%], …, (90%,100%]. Accordingly, a suffix of "4" represents a sample with a mean FSC of (40%,50%], corresponding to a mean value of 47% in this example.





Figure 7 illustrates the spatial patterns of the target FSC and its 20 associated features for this example sample. The close spatial correspondence between FSC and key physical drivers such as elevation, LST, and NDSI demonstrates strong physical coherence, reflecting how snow distribution is governed by terrain, temperature, and surface spectral properties. This internal consistency among feature variables and FSC confirms the physical realism and internal coherence of the dataset. Detailed descriptions and metadata specifications are provided in the supplementary documentation accompanying the dataset release.

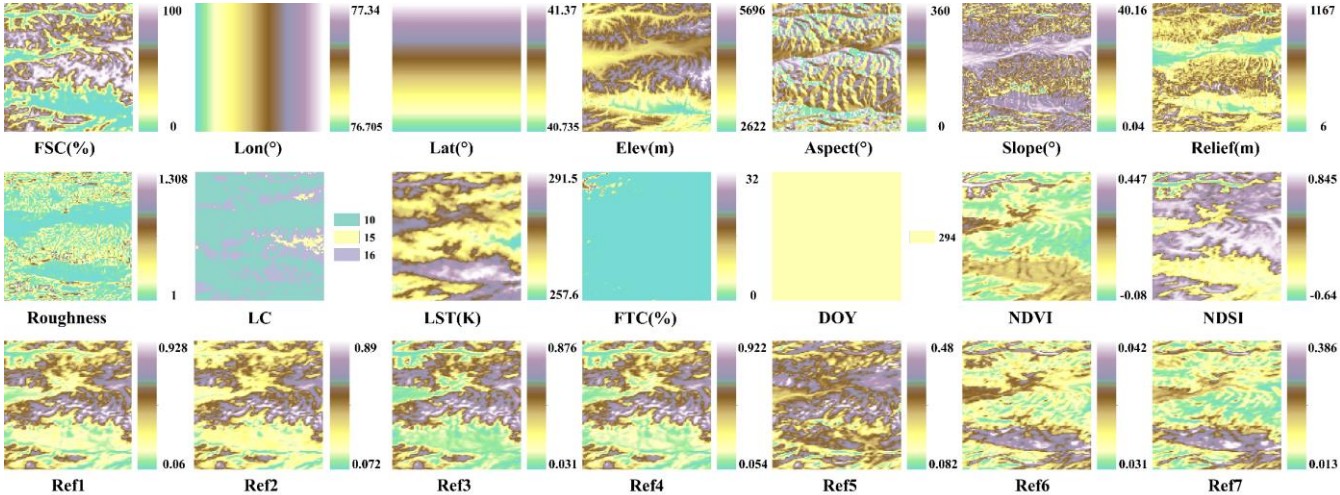


**Figure 7: The spatial patterns of the FSC and corresponding 20 features for example sample "20201020_r37c06_4"**

### 3.3.3 AI-Readiness Evaluation Protocol

To rapidly assess the AI-readiness of Earth observation datasets, the U.S. National Oceanic and Atmospheric Administration (NOAA) proposed a four-tier maturity model (Levels 0-3), corresponding to *Not AI-Ready*, *Minimal*, *Intermediate*, and
*Optimal* readiness, respectively (Christensen, 2020). This model assesses datasets from three perspectives, i.e., *data consistency, data accessibility, and metadata*, clearly defines the characteristics of each maturity level, providing a concise yet effective framework for AI-readiness assessment.

Evidently, our AI-Ready MODIS FSC dataset demonstrably achieves the *highest maturity level (Level 3)* across all dimensions. In terms of data consistency, it incorporates a fully automated internal consistency-checking and reporting mechanism.
Regarding data accessibility, it supports both traditional data download and Data-as-a-Service (DaaS) access via cloud-based and high-performance computing environments, complemented by open-source preprocessing scripts and example Python codes (via https://github.com/houjin0503/AI-Ready-China-FSC). In terms of metadata, it offers standardized, machine-readable documentation compliant with international interoperability standards.

However, NOAA's model provides only a generalized evaluation scheme and does not fully capture the multi-dimensional
characteristics of AI-ready geospatial datasets, particularly those involving multi-source harmonization, hierarchical data structures, and algorithmic usability. To address these limitations, we refined and extended NOAA's framework by introducing a comprehensive "Four Layers-Four Domains-Fifteen Attributes" (4L-4D-15A) Evaluation Protocol (Table 6).



This refined framework enables a granular, multi-perspective evaluation of AI-readiness, systematically characterizing the dataset across four complementary dimensions, i.e., *Data, Information, System, and Application*. The fifteen attributes provide
a structured and quantitative basis for evaluating usability, scalability, interoperability, and sustainability in AI-driven snow-monitoring applications. Overall, the constructed AI-Ready MODIS FSC dataset satisfies all fifteen attributes at a high maturity level, with only minor aspects, such as deeper DaaS integration and long-term sustainability management, identified for continuous improvement. Consequently, the dataset represents an optimal-level (Level 3) AI-Ready product, supporting seamless integration into machine learning, data assimilation, and large-scale Earth system modelling workflows. Beyond this
specific application, the proposed *4L-4D-15A Evaluation Protocol* offer a generalized and transferable framework that can be applied to other Earth observation datasets, providing a unified and standardized foundation for evaluating the AI-readiness, usability, and scientific reliability of geospatial data products.

**Table 6: AI-Ready data characterization evaluation protocol of "Four Layers-Four Domains-Fifteen Attributes" (4L-4D-15A).**

| Layer | Domain | Attribute | Core Description |
|---|---|---|---|
| I. Data | Data Engineering | 1. Preparation & Cleaning | Pre-cleaned, standardized, and ready for direct AI workflows. |
| | | 2. Multi-source Integration | Integrated from multi-source and multi-modal datasets with unified spatiotemporal alignment. |
| | | 3. Structure & Format | Structured and standardized formats for efficient AI processing. |
| II. Information | Information Integrity | 4. Metadata & Annotation | Comprehensive metadata and semantically consistent annotations. |
| | | 5. Quality & Integrity | Ensures spatial-temporal-physical consistency with rigorous QC. |
| | | 6. Provenance & Traceability | Full provenance records for transparency and reproducibility. |
| | | 7. Timeliness | Frequently updated and temporally consistent. |
| III. System | System Interoperability | 8. Accessibility | Accessible via APIs or open data services. |
| | | 9. Interoperability | Compliant with FAIR and OGC standards for system interoperability. |
| | | 10. Scalability | Designed for scalable and distributed AI computation. |
| | | 11. Reusability | Accompanied by clear documentation and reuse licensing. |
| IV. Application | AI Adaptability & Ethics | 12. AI-task Adaptability | Optimized for AI tasks (classification, regression, segmentation) with balanced samples. |
| | | 13. Computational Efficiency | Optimized for HPC and GPU-based AI processing. |
| | | 14.Privacy & Ethics Compliance | Compliant with data privacy and ethical standards. |
| | | 15. Sustainability & Maintenance | Version-controlled and maintained for long-term sustainability. |

**3.4 Dataset Organization and Partitioning**

The AI-Ready China FSC dataset is hierarchically organized to support both temporal and spatial generalization in AI-based modelling. Spatially, it is divided into four subregional domains, XJ, NEIM, TP, and Other, representing the major snow-climate zones across mainland China. Within each subregion, samples are partitioned into training, validation, and testing subsets following a 2:1:1 spatial ratio (Fig. 8), ensuring balanced geographic representation and minimizing spatial autocorrelation among subsets. This spatial independence is essential for robust model evaluation, guaranteeing that
performance metrics reflect true generalization rather than local dependence. Temporally, each spatial subset is organized into 22 snow seasons (2000-2001, 2001-2002, …, 2021-2022), where each season directory contains separate FSC and Slice folders for FSC reference files and corresponding feature files, respectively. This nested spatial-temporal architecture preserves both



Earth System
Science
Data

the geographical heterogeneity and the chronological continuity of snow processes, enabling season-specific analysis and multi-year learning. The resulting spatial-first and temporally nested design achieves an optimal balance between regional

representativeness, computational efficiency, and interannual comparability. It facilitates cross-validation, inter-seasonal benchmarking, and domain adaptation, supporting AI applications from local retrieval to continental-scale modelling. Furthermore, the standardized directory structure enhances usability, transparency, and reproducibility, allowing users to efficiently locate, trace, and aggregate samples across regions and years. Collectively, this structured organization not only ensures reproducible AI workflows but also provides a scalable blueprint for future AI-ready Earth observation datasets.

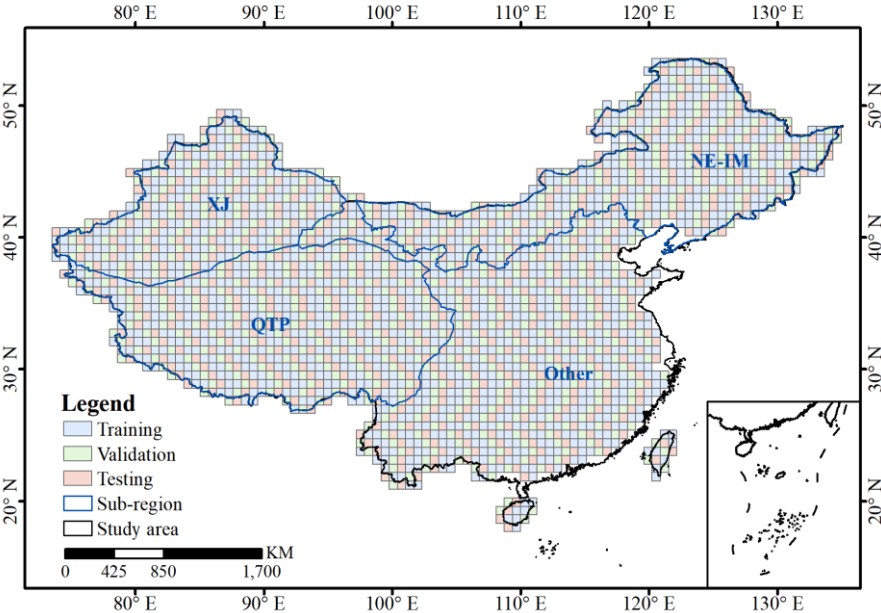


**Figure 8: Spatial distribution of samples assigned to training, calibration, and testing dataset with a 2:1:1 ratio across the study area.**

## 4. Demonstration Applications

To demonstrate the quality, reliability, and applicability of the AI-Ready FSC dataset, three representative applications were conducted: (1) benchmark modelling using relatively simple, well-established algorithms without extensive hyperparameter

tuning to evaluate dataset robustness and usability, (2) assessment of MODIS FSC product accuracy using the dataset, and (3) large-scale seamless FSC mapping over China to examine the dataset's representativeness across the study region and the generalization ability of AI models. These experiments aimed to provide a transparent and robust baseline rather than optimized prediction performance.



## 4.1 Benchmark Modelling

To evaluate the quality, reliability, and applicability of the AI-Ready FSC dataset, a set of benchmark models was established using samples from the 2021-2022 snow season. Six representative algorithms, i.e., ANN, SVR, RF, CNN, UNet, and ResNet, were implemented to assess the dataset's performance across different modelling paradigms. Each model was trained, validated, and tested using spatially independent data splits derived from the AI-Ready sample structure, thereby ensuring that the performance evaluation reflects the dataset's capability for geographic generalization rather than random sample fitting. Model

performance was assessed using three standard metrics: root mean square error (RMSE), mean absolute error, and coefficient of association (R). The detailed architectural configurations, hyperparameter settings, and training strategies for all benchmark models are summarized in Table 7, providing a reproducible foundation for model replication and comparative analysis. It is important to emphasize that all benchmark models were implemented using their canonical architecture and commonly adopted hyperparameter settings, without any deliberate optimization of network structure, parameter tuning, or learning algorithm

modification. This design choice ensures that the benchmarking process isolates and highlights the intrinsic quality, consistency, and representativeness of the AI-Ready FSC samples, rather than reflecting model-specific tuning effects. Accordingly, the resulting performance metrics offer an objective and unbiased evaluation of the dataset's robustness, reliability, and applicability for diverse AI-driven snow-cover modelling frameworks. These results thus serve as a baseline reference for subsequent algorithmic development, dataset intercomparison, and large-scale AI-readiness benchmarking within

the cryosphere research community.

**Table 7: The architectural configurations and parameter settings of the six benchmark models**

| Method | Model Structure | Model Parameters |
|--------|-----------------|------------------|
| ANN | Three hidden layers with 128, 64, and 32 neurons | Optimizer: Adam; Learning rate: 1e-3; Loss: MSE; Max epochs: 200; Early stopping patience: 20 |
| SVR | Nonlinear mapping with radial basis function (RBF) kernel | Penalty (C): 10; Kernel parameter: scale |
| RF | Ensemble of decision trees trained on bootstrap strategy, outputs are aggregated by averaging | Total trees: 200; Max features: sqrt(200); Max depth: 20; |
| CNN | 3× (Conv3×3, BatchNorm2d, ReLU) | Optimizer: Adam; Batch size: 32; Learning rate: 1e-3; Loss: MSE; Max epochs: 200; Early stopping patience: 20 |
| UNet | Three encoder layers, bottleneck, 3 decoder layers with skip connections; each conv block: 2× (Conv3×3, BatchNorm2d, ReLU) | Optimizer: Adam; Batch size: 32; Learning rate: 1e-3; Loss: MSE; Max epochs: 200; Early stopping patience: 20 |
| ResNet | Four residual blocks, each combining convolutional layers (Conv3×3, BatchNorm2d, ReLU) with skip connections each residual block | Optimizer: Adam; Batch size: 32; Learning rate: 1e-3; Loss: MSE; Max epochs: 200; Early stopping patience: 20 |

Figures 9 and 10 illustrate the FSC distributions estimated by six benchmark models for ten representative samples drawn from the training and testing datasets, respectively, spanning the full FSC range of [0-10%], [10-20%], …, [90-100%]. Overall, all models successfully capture the primary spatial patterns of snow cover, demonstrating the capability of both ML and DL

approaches to reproduce tile-scale FSC variability. Among the point-based ML models, ANN, SVR, and RF effectively approximate spatial heterogeneity in FSC, although SVR tends to exhibit slightly weaker performance, particularly in complex terrain or intermediate FSC ranges. Estimation errors generally increase in areas with extensive snow coverage, likely due to



spectral saturation, canopy occlusion, and mixed-pixel effects. In contrast, the tile-based DL models, i.e., CNN, UNet, and ResNet, demonstrate enhanced capability in capturing coherent spatial structures and snow boundaries, yielding FSC

distributions more consistent with reference values and effectively reducing local discrepancies, especially in continuous snow-covered regions. Comparative analysis between training and testing results reveals slightly higher accuracy for training samples, reflecting their stronger spatial representativeness; however, DL models maintain robust generalization and transferability across independent test regions. Collectively, these findings confirm that both point- and tile-scale AI models can effectively reproduce FSC spatial patterns, with DL architectures offering superior accuracy and spatial coherence. More importantly, the

results highlight that the constructed AI-ready FSC dataset provides a physically consistent, statistically representative, and algorithmically versatile foundation for training and evaluating AI-based snow-cover models, supporting large-scale, high-precision snow mapping across heterogeneous regions such as China.







**Figure 9: FSC distributions of the reference truth (sample labels), six benchmark models, and the MODIS FSC product for example training samples. Panels (a)-(j) represent different snow cover conditions, spanning the full FSC range of [0-10%], [10-20%], …, [90-100%], respectively.**








Figure 10: Same as Figure 9, but samples from the testing dataset.





Figure 11 presents the performance comparison of the six benchmark models on both the training and testing datasets. On the

training set, overall performance was robust, with the ResNet model achieving the best results (R = 0.89, RMSE = 11.69%, MAE = 8.13%), followed closely by UNet. On the testing set, the UNet model achieved the highest performance (R = 0.86, RMSE = 14.21%, MAE = 9.57%), slightly outperforming ResNet (R = 0.86, RMSE = 14.33%, MAE = 9.84%). The point-scale ML models (ANN, SVR, and RF) generally performed worse than the tile-scale DL models (CNN, UNet, and ResNet), although all achieved satisfactory accuracy levels. Overall, the UNet model demonstrated the best balance between predictive

accuracy and stability. A more detailed quantitative evaluation across different FSC intervals (Table 8) further supports this conclusion. Under low (FSC < 20%) or high (FSC > 70%) snow-cover conditions, the surface environment is relatively homogeneous, dominated either by bare-ground reflectance or by continuous snow-covered surfaces. This spectral uniformity minimizes within-window variability, resulting in more stable feature-response relationships and consequently higher model accuracy. In contrast, at intermediate FSC levels (20%-70%), the coexistence of snow, vegetation, and exposed soil increases

sub-pixel heterogeneity and introduces nonlinear interactions among spectral, thermal, and topographic factors. These conditions amplify model uncertainty and lead to a noticeable decrease in prediction accuracy for all models. In short, the results indicate that model performance is governed more by the quality, representativeness, and internal consistency of the training data than by the specific model architecture. When the training samples adequately capture the full range of spectral, topographic, and environmental variability, even relatively simple models can achieve high predictive accuracy. Conversely,

insufficient or biased samples can limit the learning capacity of advanced architecture. This finding underscores that the intrinsic data quality and representativeness of the AI-ready FSC samples are the primary determinants of model generalization and robustness.

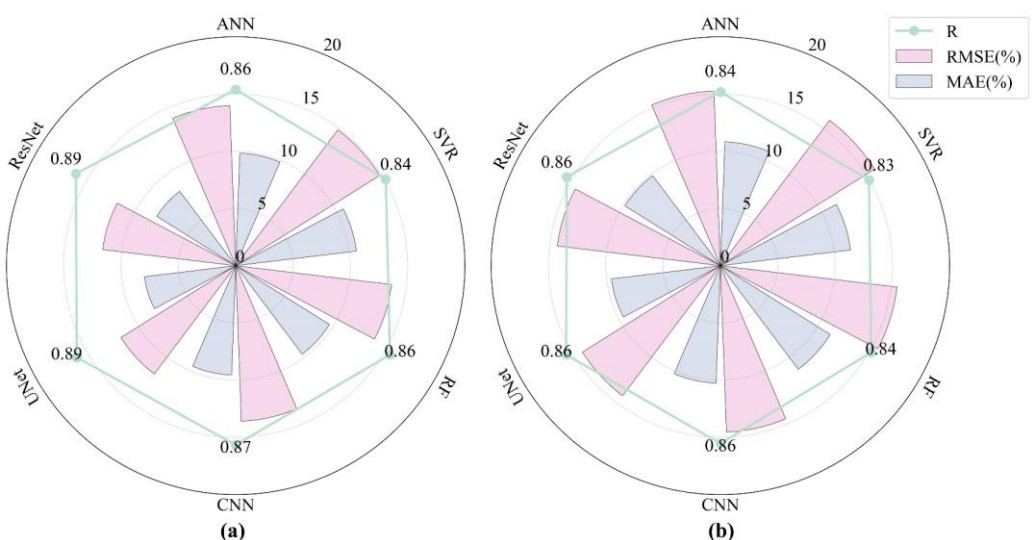

**Figure 11: Performance comparison of six benchmark models on the training and testing FSC sample datasets. Panels (a) and (b)**
**show the evaluation results for the training and testing subsets, respectively.**

**Table 8. Quantitative evaluation of six benchmark models and standard MODIS FSC product based on different FSC intervals**





|  | FSC intervals | ANN | SVR | RF | CNN | UNet | ResNet | MODIS |
|---|---|---|---|---|---|---|---|---|
| **RMSE (%)** | [0,10] | 9.93 | 10.20 | 9.55 | 9.24 | 8.44 | 8.47 | 8.99 |
| | [10,20] | 12.45 | 12.78 | 12.04 | 11.81 | 11.13 | 11.14 | 14.01 |
| | [20,30] | 15.00 | 15.70 | 15.00 | 14.59 | 13.59 | 13.64 | 15.35 |
| | [30,40] | 16.51 | 17.41 | 16.66 | 15.98 | 14.79 | 14.76 | 15.91 |
| | [40,50] | 17.56 | 18.55 | 17.78 | 17.06 | 15.67 | 15.70 | 16.51 |
| | [50,60] | 15.99 | 16.63 | 16.13 | 15.82 | 14.89 | 14.80 | 16.85 |
| | [60,70] | 14.59 | 15.27 | 14.56 | 13.65 | 12.95 | 12.90 | 15.66 |
| | [70,80] | 14.97 | 15.69 | 14.93 | 13.93 | 13.34 | 13.30 | 14.12 |
| | [80,90] | 15.01 | 15.98 | 14.97 | 14.21 | 12.61 | 12.61 | 13.92 |
| | [90,100] | 14.30 | 15.59 | 14.71 | 14.00 | 11.05 | 11.41 | 13.76 |
| | Average | 14.63 | 15.38 | 14.63 | 14.03 | 12.85 | 12.87 | 14.51 |
| **MAE (%)** | [0,10] | 6.72 | 7.04 | 6.50 | 5.91 | 4.82 | 5.21 | 6.04 |
| | [10,20] | 8.49 | 8.82 | 8.19 | 7.90 | 6.89 | 7.19 | 9.64 |
| | [20,30] | 10.82 | 11.30 | 10.92 | 10.39 | 9.29 | 9.49 | 10.68 |
| | [30,40] | 12.06 | 12.71 | 12.33 | 11.66 | 10.44 | 10.54 | 11.37 |
| | [40,50] | 12.99 | 13.73 | 13.32 | 12.61 | 11.25 | 11.38 | 12.10 |
| | [50,60] | 11.48 | 12.10 | 11.78 | 11.41 | 10.52 | 10.60 | 11.96 |
| | [60,70] | 10.12 | 10.91 | 10.33 | 9.63 | 8.95 | 9.03 | 11.20 |
| | [70,80] | 9.99 | 10.97 | 10.29 | 9.55 | 8.91 | 9.01 | 9.95 |
| | [80,90] | 9.91 | 11.23 | 10.27 | 9.67 | 8.33 | 8.51 | 10.45 |
| | [90,100] | 9.36 | 11.09 | 10.07 | 9.65 | 7.18 | 7.69 | 10.98 |
| | Average | 10.19 | 10.99 | 10.40 | 9.84 | 8.66 | 8.87 | 10.44 |
| **R** | [0,10] | 0.82 | 0.82 | 0.84 | 0.85 | 0.86 | 0.86 | 0.83 |
| | [10,20] | 0.84 | 0.83 | 0.85 | 0.85 | 0.87 | 0.87 | 0.80 |
| | [20,30] | 0.85 | 0.83 | 0.85 | 0.85 | 0.88 | 0.87 | 0.86 |
| | [30,40] | 0.85 | 0.83 | 0.85 | 0.86 | 0.88 | 0.88 | 0.89 |
| | [40,50] | 0.85 | 0.84 | 0.85 | 0.86 | 0.88 | 0.88 | 0.89 |
| | [50,60] | 0.88 | 0.87 | 0.88 | 0.89 | 0.90 | 0.90 | 0.90 |
| | [60,70] | 0.89 | 0.88 | 0.89 | 0.90 | 0.91 | 0.91 | 0.89 |
| | [70,80] | 0.84 | 0.83 | 0.84 | 0.86 | 0.87 | 0.87 | 0.87 |
| | [80,90] | 0.76 | 0.74 | 0.76 | 0.77 | 0.81 | 0.81 | 0.76 |
| | [90,100] | 0.64 | 0.63 | 0.66 | 0.67 | 0.75 | 0.74 | 0.66 |
| | Average | 0.91 | 0.90 | 0.91 | 0.92 | 0.93 | 0.93 | 0.92 |

## 4.2 Standard MODIS FSC Product Accuracy Assessment

Furthermore, all reference FSC values from the 2021-2022 snow season were employed to independently validate the accuracy of the standard MODIS FSC product. Following the official retrieval algorithm proposed by Salomonson and Appel (2004), FSC estimates were derived from the NDSI values in the MOD10A1 product. As summarized in Table 8, the quantitative comparison shows that the standard MODIS FSC product achieves reasonable accuracy, with performance comparable to the point-scale ML models (ANN, SVR, and RF). However, it remains clearly inferior to the tile-scale DL models (CNN, UNet, and ResNet), highlighting that spatially explicit architectures can more effectively capture contextual information and spatial continuity in FSC estimation. Moreover, since the MODIS standard algorithm retrieves FSC only under clear-sky conditions, the average effective coverage is typically below 50%, indicating that more than half of the surface area is cloud-obscured and thus excluded from estimation. This limitation substantially restricts its spatial completeness. In contrast, the AI-Ready FSC sample dataset constructed in this study exhibits robust performance under diverse atmospheric and surface conditions, ensuring high spatial and temporal continuity. The FSC distribution maps for representative sample tiles (Fig. 9 and 10) further





support these findings, visually confirming the superior consistency and realism of the AI-based FSC estimates. Importantly,
the large-scale validation involving 3686 independent samples, each corresponding to one Landsat or Sentinel-2 scene, ensures
the statistical robustness and credibility of the evaluation, providing strong evidence for the high quality, reliability, and
representativeness of the constructed dataset.

**4.3 Large-Scale FSC Mapping across China**

Figure 12 illustrates the spatial FSC distribution across China on three representative dates, i.e., October 1, 2021 (accumulation
period), January 18, 2022 (stable period), and March 8, 2022 (ablation period), as estimated by the Unet model developed in
Section 4.1. The retrieved patterns exhibit clear spatial continuity and physically consistent gradients with respect to both
latitude and elevation. During the early accumulation phase (October 1, 2021), snow cover appears primarily over high-altitude
regions, including the QTP, the Tianshan and Altai Mountains, and the northeastern highlands (Greater Khingan Range and
Changbai Mountains), while most lowlands and southeastern coastal regions remain snow-free. By January 18, 2022, the snow
extent reaches its annual maximum, forming an almost continuous snow belt across the QTP and northern China, with FSC
values exceeding 0.8 in cold and high-elevation zones. On March 8, 2022, a clear retreat of snow cover is observed,
characterized by rapid melting in low- and mid-latitude areas, whereas residual snow persists in high mountains and northern
forests, depicting a physically consistent seasonal evolution.

Spatially, the Unet-derived FSC maps exhibit smooth transitions and coherent snowline boundaries, indicating that the model
effectively captures contextual terrain information and suppresses pixel-level noise. The altitude-FSC relationship remains
nearly monotonic, with FSC increasing systematically with elevation, further validating the physical realism of the estimates.
Minor discontinuities and isolated patches occur mainly along steep slopes or shaded terrains, where topographic shadows or
mixed-pixel effects may distort spectral responses. When compared with MODIS clear-sky FSC observations, the Unet results
display strong spatial agreement across most regions. Based on two randomly selected representative tiles for each date (Table
9), the UNet-derived FSC achieves mean R above 0.92, with an average RMSE of 10.75% and MAE of 6.46%, notably
outperforming the corresponding MODIS product (RMSE=4.41%, MAE=9.34%, R= 0.91). Although slight biases persist,
such as minor underestimation in high-FSC areas and overestimation in low-FSC zones, the UNet model demonstrates clear
advantages in spatial continuity, snowline delineation, and temporal consistency. Overall, these results confirm that the Unet-
based FSC mapping provides physically reliable, spatially coherent, and temporally consistent characterization of snow-cover
dynamics across China. More importantly, the constructed AI-Ready FSC sample dataset establishes a robust, standardized,
and scalable foundation for training, validating, and benchmarking advanced AI models, thereby enabling high-precision,
large-scale snow-cover mapping over complex and heterogeneous regions such as China.




**Figure 12: Spatial distribution of FSC across China on three representative dates derived from the standard MODIS FSC product**
**(subplots (a), (c), and (e)) and the UNet model estimates (subplots (b), (d), and (f)). Panels (a)-(b) correspond to October 1, 2021**
**(accumulation period), (c)-(d) to January 18, 2022 (stable period), and (e)-(f) to March 8, 2022 (ablation period). The UNet-derived**
**FSC maps exhibit improved spatial continuity and physically consistent gradients relative to the MODIS product. Insets show**
**example tiles, i.e., r30c28, r25c28, r42c62, r27c38, r27c41, and r25c40 (from top to bottom), highlighting local-scale improvements**
**in snow delineation and spatial coherence.**

**Table 9. Quantitative evaluation of randomly selected example samples on three representative dates, i.e., October 1, 2021**
**(accumulation period), January 18, 2022 (stable period), and March 8, 2022 (ablation period), from UNet model and standard**
**MODIS FSC product.**

| Model | Sample (tile) | Reference FSC (%) | RMSE (%) | MAE (%) | R |
|---|---|---|---|---|---|
| **UNet** | r30c28 | 6.31 | 5.89 | 2.74 | 0.97 |
| | r25c28 | 32.08 | 8.01 | 5.18 | 0.98 |
| | r42c62 | 26.22 | 16.38 | 7.82 | 0.94 |
| | r27c38 | 22.77 | 10.94 | 6.38 | 0.97 |
| | r27c41 | 49.38 | 10.44 | 7.81 | 0.94 |
| | r25c40 | 44.81 | 12.85 | 8.85 | 0.93 |
| **MODIS FSC** | r30c28 | 6.31 | 7.63 | 3.42 | 0.96 |
| | r25c28 | 32.08 | 12.81 | 7.33 | 0.96 |
| | r42c62 | 26.22 | 17.24 | 9.34 | 0.92 |
| | r27c38 | 22.77 | 12.54 | 8.03 | 0.95 |
| | r27c41 | 49.38 | 18.28 | 14.28 | 0.90 |
| | r25c40 | 44.81 | 17.98 | 13.64 | 0.91 |





## 5. Discussion

### 5.1 Potential Applications of the ChinaAI-FSC Dataset

Although this study focuses on a single snow season (2021-2022) to demonstrate the use of the ChinaAI-FSC dataset through benchmark modelling, MODIS FSC product evaluation, and large-scale FSC mapping. Its standardized structure, multi-source features, and high-quality reference samples hold far broader potential applications across snow, cryosphere, and AI research domains.

**(1) AI-based FSC Retrieval and Model Benchmarking**

ChinaAI-FSC provides a standardized foundation for developing, optimizing, and evaluating AI algorithms at local and national scales. It enables fair, reproducible comparisons among various ML/DL models for FSC estimation under different environmental and snow conditions.

**(2) Evaluation of Multi-source Snow Products**

The Landsat/Sentinel-2 derived high-resolution reference FSC can serve as an independent benchmark for assessing the
accuracy of MODIS, VIIRS, and future snow products over the past two decades across China

**(3) Enhancement of Standard MODIS Snow Products**

The dataset supports the improvement of standard MODIS snow products by providing AI-ready reference samples for cloud removal, bias correction, and temporal gap-filling. These applications can help reconstruct seamless, long-term FSC records and improve product accuracy and consistency.

**(4) Long-Term and Seamless FSC Product Generation**

By integrating multi-source features and high-quality reference FSCs, the dataset can facilitate large-scale and temporally continuous FSC product generation over China, extending to multi-year or decadal scales for climate and hydrological analysis.

**(5) Transfer Learning for Global FSC Estimation**

The large-scale, diverse samples enable investigation of model transferability to other snow-covered regions across the
Northern Hemisphere, contributing to the development of globally consistent AI-based FSC estimation frameworks.

**(6) Feature Optimization and Model Interpretability**

With 20 feature variables, the dataset allows systematic feature selection and sensitivity analysis to identify redundant or less influential features, improving model efficiency, interpretability, and generalization.

**(7) Multi-scale Spatial Modelling**

Each sample tile (128 × 128 MODIS pixels) supports both point-scale and tile-scale modelling with varying sizes (e.g., 8×8, 16×16, 32×32, or full 128×128 tiles), enabling evaluation of spatial context effects on FSC estimation and exploring the advantages of neighbourhood-based DL architectures.

**(8) Physics-AI Hybrid Modelling**

The dataset provides opportunities to integrate AI approaches with physical snow models, advancing physics-informed or
constraint-guided AI frameworks for improved snow process representation.





**(9) Data Assimilation and Earth System Model Integration**

Beyond static FSC mapping, the dataset can support data assimilation experiments within land surface, hydrological, and climate models. AI-derived FSC estimates can be assimilated as observational constraints to improve model initialization, snowpack evolution simulation, and runoff forecasting. Such integration strengthens the coupling between AI-based data
products and process-based Earth system simulations, enhancing representations of cryosphere-hydrosphere-atmosphere interactions.

**(10) Open Science and Community Benchmarking**

As an open-access, FAIR-compliant dataset, ChinaAI-FSC promotes community-driven model comparison, continuous dataset refinement, and transparent research practices, ensuring long-term value for cryosphere and Earth system science.
Overall, the ChinaAI-FSC dataset provides a flexible and extensible foundation for AI-based snow research. Its potential applications, from benchmarking to hybrid modelling, highlight its value for advancing data-driven and physically consistent snow monitoring at regional to continental scales.

**5.2 Limitations and Future Work**

The ChinaAI-FSC dataset provides a robust foundation for developing and evaluating advanced AI-based FSC estimation
models, enabling reproducible benchmarking, cross-regional analyses, and methodological innovation in snow-related remote sensing research. Despite these strengths, several limitations remain that merit consideration in future work.

A primary limitation concerns sample balance across snow conditions and geographic regions. Although the dataset covers diverse terrain types, elevation gradients, and climatic zones across China, certain snow regimes, particularly extremely sparse or extremely dense snow cover, are relatively underrepresented. Such imbalance may bias model training and limit
generalization to rare or edge-case scenarios. Future efforts will address this by systematically augmenting underrepresented snow conditions, employing adaptive sampling strategies, physically consistent data augmentation, and targeted acquisition of high-quality reference imagery to improve the statistical and environmental balance of samples.

Another limitation involves the current spatial extent, which is restricted to mainland China. Extending the dataset to global or hemispheric scales is crucial for promoting AI model transferability and enabling global snow monitoring applications.
Achieving this goal will require integrating multi-sensor satellite imagery (e.g., Sentinel-2, Landsat-8/9, VIIRS, and MODIS), harmonizing data across diverse land-cover and terrain conditions, and establishing cross-sensor calibration and interoperability protocols to ensure spatiotemporal consistency. Expanding the dataset's geographic scope will further support comparative analyses of snow dynamics under different climatic regimes, enhancing its scientific value for global cryosphere and hydrological research.

Feature diversity and temporal consistency represent additional areas for improvement. While the current dataset includes 20 feature variables encompassing spectral, topographic, and environmental factors, future versions could incorporate auxiliary datasets such as snow albedo, precipitation, soil moisture, and vegetation phenology. This expanded feature space would enable AI models to better characterize complex snow-terrain-climate interactions. Maintaining temporal continuity and calibration





consistency across multiple snow seasons will also be essential for developing long-term, stable FSC products, thereby facilitating research on interannual variability and climate-driven snow cover trends.

Uncertainty in the dataset primarily stems from cloud contamination, terrain shadowing, and sensor discrepancies, which can affect FSC retrieval accuracy in specific regions or during certain seasons. Mitigating these uncertainties requires advances in cloud masking algorithms, topographic correction methods, and cross-sensor harmonization. Furthermore, integrating uncertainty quantification frameworks, such as probabilistic modelling, ensemble prediction, or Bayesian error propagation, can enhance model reliability and provide users with confidence estimates alongside FSC predictions.

Data preparation should also be viewed as an iterative process. As new satellite observations become available and AI models continue to evolve, periodic updates to the dataset and its preparation protocols will be essential to maintain relevance, accuracy, and scalability. In summary, although the ChinaAI-FSC dataset already represents a major step forward in AI-driven snow cover research, further improvements in sample balance, spatial coverage, feature enrichment, uncertainty management, and continuous updating will be critical to its future development. These enhancements will expand its applicability, promote reproducible and robust AI modelling, and strengthen its utility as a versatile foundation for regional and global snow monitoring.

## 6 Code and data availability

The ChinaAI-FSC dataset (Hou et al., 2025) is publicly available at the National Tibetan Plateau Data Center (TPDC) at https://doi.org/10.11888/Cryos.tpdc.303034 (also accessible via https://cstr.cn/18406.11.Cryos.tpdc.303034) and from Zenodo at https://doi.org/10.5281/zenodo.17707386, hosted on an open-access repository that supports persistent identifiers and version control. The full dataset, including feature variable tiles, reference FSC tiles, metadata files, and documentation, can be accessed and downloaded under a Creative Commons Attribution (CC BY 4.0) license.

All associated code, including data reading examples, processing scripts, and benchmark modeling workflows, is publicly available in our GitHub repository: https://github.com/houjin0503/AI-Ready-China-FSC. This ensures full transparency, traceability, and reproducibility of both the data generation and modelling processes.

## 7 Summary

This study presents ChinaAI-FSC, the first large-scale, AI-ready MODIS FSC sample dataset for mainland China, establishing a standardized, high-quality foundation for AI-based snow monitoring and modelling. The dataset integrates multi-source satellite observations with rigorous physical and quality control procedures, while implementing a novel "Four Layers-Four Domains-Fifteen Attributes" (4L-4D-15A) evaluation framework to ensure spatiotemporal representativeness, physical consistency, environmental completeness, and metadata standardization.

By fully embodying the AI-readiness paradigm, ChinaAI-FSC provides a reproducible and interoperable basis for the development, benchmarking, and intercomparison of ML and DL models for FSC estimation. It enables large-scale and temporally consistent FSC mapping, facilitates cross-sensor validation (e.g., between MODIS and VIIRS), and supports multi-scale AI workflows from local retrieval to continental-scale modelling. Beyond serving as a benchmark dataset, ChinaAI-FSC also contributes to methodological innovation in several key areas: (i) feature optimization and AI model interpretability, by providing richly annotated feature-response pairs suitable for sensitivity and explainability analyses; (ii) physics-AI hybrid modelling, by offering high-quality training and validation data that enable the integration of physical constraints into data-driven frameworks; and (iii) data assimilation and Earth system modelling, by supplying consistent observational inputs for model calibration and coupling.

While current limitations remain, such as imbalanced snow-condition samples, restricted spatial coverage over China, and residual uncertainties linked to cloud contamination, complex terrain, and sensor discrepancies, these also define clear pathways for future enhancement. Upcoming releases will focus on extending spatiotemporal coverage, enriching feature diversity, and strengthening cross-sensor harmonization to further improve dataset completeness and continuity. Overall, ChinaAI-FSC represents a versatile, open, and FAIR-compliant resource that advances AI-driven snow monitoring and model development, enhances algorithmic robustness and interpretability, and supports regional-to-global assessments of cryosphere dynamics under a rapidly changing climate.

## Funding

This work was supported by National Natural Science Foundation of China (Grant No. 42130113, 42371398, 42471434, and 42361060), and the program of the Key Laboratory of Cryospheric Science and Frozen Soil Engineering, CAS (No. CSFSE-ZZ-2409).

## Competing interests

The contact author has declared that none of the authors has any competing interests.

## Author contribution

JH and CH conceived the study and designed the overall methodology. JH, MZ, and YZ developed the model code and conducted the simulations. XH, JG, and PD performed the formal analyses. CH provided key resources and secured project funding. JH prepared the original draft of the manuscript. All authors contributed to the review and editing of the final manuscript.



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
