# Peer review of "ChinaAI-FSC: A Comprehensive AI-Ready MODIS Fractional Snow Cover Dataset for China (2000-2022)"

_Earth System Science Data, 2025_

## Author Comment (AC1)

**Response to Reviewer 1**

We greatly appreciate the reviewer's insightful and constructive comments, which have significantly helped to improve the quality and clarity of this manuscript. The manuscript has been carefully revised accordingly. The reviewer's comments are shown in **black**, and our responses are provided in **blue**.

This manuscript presents the development and evaluation of the ChinaAI-FSC dataset, a comprehensive AI-ready MODIS fractional snow cover sample collection for China spanning 2000-2022. The objective of this work is to provide a standardized, large-scale, and high-quality benchmark for AI-driven snow cover mapping. The authors have undertaken a substantial effort in data integration, quality control, and validation, and the introduction of a novel "4L-4D-15A" evaluation framework is a notable strength. However, the manuscript in its current form has several issues that need to be justified. The most critical concerns revolve around the potential imbalance of samples across varying snow conditions and geographic regions, as well as insufficient discussion regarding the sources and mitigation of uncertainty. These aspects affect the perceived robustness and broad applicability of the dataset and must be thoroughly addressed before publication.

**Response:**

We sincerely thank the reviewer for the careful reading of our manuscript and for the insightful comments. We agree with the concerns regarding sample imbalance and uncertainty, which are indeed important for assessing the applicability of AI-ready FSC datasets. In response to these concerns, we have carefully revised the manuscript following these suggestions.

Regarding sample imbalance, we have revised the manuscript to explicitly acknowledge and discuss this issue in Section 5.2 (Data availability constraints and sample imbalance). We clarify that the construction of ChinaAI-FSC is fundamentally constrained by the availability of near-cloud-free, high-quality Landsat and Sentinel-2 observations, which are inherently uneven in space and time, especially over mountainous and persistently cloudy regions. Consequently, the dataset was assembled by collecting all available reference-quality observations that met strict quality criteria, rather than by imposing an explicit sampling strategy aimed at balancing snow conditions. As a result, some imbalance remains, particularly for extremely sparse and extremely dense snow cover fractions.

We also clarify that such imbalance reflects the intrinsic spatiotemporal characteristics of snow cover and the observational limitations of optical remote sensing, rather than deficiencies in the dataset construction methodology. To assess whether this imbalance critically limits AI-based FSC modelling, we conducted baseline experiments using simple benchmark models without specialized architectures or extensive parameter tuning. These tests indicate that, even under the existing sample distribution, AI-based FSC estimation can achieve satisfactory performance, suggesting that the dataset already provides meaningful learning signals. At the same time, we note that incorporating more advanced sample representativeness analysis techniques (e.g., hierarchical clustering, density-based selection, or reweighting strategies) may further improve model performance, and we explicitly frame ChinaAI-FSC as a reference dataset that enables such methodological investigations.

Regarding uncertainty, we have substantially expanded the discussion in Section 5.3 to systematically identify major uncertainty sources, including cloud contamination, terrain-induced illumination effects, forest canopy obscuration, mixed-pixel conditions, and cross-sensor inconsistencies. We explain how these uncertainties propagate into both FSC reference estimates and predictor variables, particularly in complex terrain and forested environments. While quality control and physically consistent feature selection help mitigate these effects, we acknowledge that residual uncertainties are unavoidable in optical snow remote sensing. We further discuss potential mitigation pathways, including uncertainty-aware modelling, ensemble learning, and probabilistic or Bayesian frameworks, to provide users with a clearer understanding of dataset limitations and appropriate usage.

Overall, the revised manuscript places greater emphasis on transparently characterizing sample imbalance and uncertainty, rather than minimizing or obscuring them. We hope that these revisions clarify the realistic scope, limitations, and methodological value of the ChinaAI-FSC dataset, and address the reviewer's concerns regarding its applicability and scientific grounding.

**Major Comments:**

What is the innovative aspect of this article? Is it a technological innovation or a methodological innovation?

**Response:**

Thank you for raising this important point. We clarify that the primary innovation of this study is **methodological rather than algorithmic**.

At present, there is no AI-ready FSC sample repository at either regional or hemispheric scale, nor a standardized and reproducible framework for constructing and validating FSC datasets explicitly designed for machine learning and deep learning. Existing FSC studies typically rely on locally collected samples or product-level validations, which are highly heterogeneous in reference generation, quality control, and data organization, thereby limiting reproducibility, cross-regional model generalization, and fair algorithm comparison.

The core innovation of this work is the establishment of a continental-scale AI-ready FSC dataset paradigm, together with a transparent construction and evaluation framework. Specifically, the methodological contributions include:

(1) A continental-scale AI-ready FSC sample repository

We construct ChinaAI-FSC over mainland China, a region encompassing most major Northern Hemisphere snow regimes, providing a representative and standardized benchmark that bridges the gap between local experimental datasets and global product-level archives.

(2) A complete and transparent AI-ready dataset construction workflow

We develop a transparent workflow covering high-resolution reference FSC generation, multi-source feature extraction, feature-target matchup construction, pixel- and tile-level consistency-based quality control, and standardized spatial tiling (128×128 MODIS pixels), producing learning-ready samples without ad-hoc preprocessing.

(3) Consistency-based multi-layer quality control

A pixel- and tile-level screening strategy is implemented to reduce unreliable or internally inconsistent samples, improving the robustness and generalizability of AI model training.

(4) A formal AI-readiness evaluation framework (4L-4D-15A)

We introduce an FSC-oriented extension of NOAA's AI maturity model to systematically assess dataset-level AI-readiness across multiple dimensions.

To avoid any ambiguity, we have revised the Introduction and Conclusion to explicitly state that the novelty of this work lies in dataset methodology, standardization, and AI-readiness engineering, rather than in proposing a new FSC retrieval algorithm.

The statements regarding the dataset's utility appear overstated or misaligned with its actual characteristics as presented. The authors should modify these claims to accurately reflect the dataset's demonstrated strengths and limitations.

**Response**:

We appreciate this important comment. We agree that some statements in the original manuscript regarding the utility of the ChinaAI-FSC dataset could be interpreted as broader than what is directly demonstrated. In response, we have revised the manuscript to ensure that all claims are grounded in experimental evidence and clearly acknowledge the dataset's limitations.

In particular, the final paragraph of the Introduction has been revised to better align with the analyses presented in the paper. The revised text now emphasizes that ChinaAI-FSC is a standardized and AI-ready MODIS FSC dataset for China, spanning 22 snow seasons, and highlights the methodological focus on constructing, validating, and evaluating FSC datasets suitable for large-scale machine learning and deep learning applications, rather than promoting speculative applications. This revision clarifies the dataset's purpose and positions it within a reproducible workflow, including feature-FSC matching, consistency-based quality control, sample quality assessment, and the AI-readiness evaluation framework (4L-4D-15A).

Additionally, Section 5.1 "Potential Applications of the ChinaAI-FSC Dataset" has been replaced with "Methodological Implications of AI-Ready FSC Dataset Construction", which emphasizes learning validity, physical consistency, and structural coherence in dataset construction rather than any untested downstream applications. The revised manuscript also clarifies key limitations related to observational constraints, sample imbalance, and residual uncertainties arising from subpixel heterogeneity, terrain effects, and canopy occlusion (as shown in new section 5.2 and 5.3). By making these aspects transparent, the dataset supports uncertainty-aware modelling and provides a reproducible, methodologically rigorous foundation for AI-driven FSC estimation.

Finally, while the current implementation focuses on snow cover, the introduced AI-readiness evaluation framework provides a transferable methodological reference for constructing and assessing AI-ready geophysical datasets more broadly.

Among these samples, what is the proportion of completely snow-free cases? If such samples account for an excessively high percentage, their contribution to research on fractional snow cover estimation may be limited.
**Response**:

Thank you for this important question. We acknowledge that the original manuscript did not explicitly report the proportion of completely snow-free samples, which is an important aspect for interpreting the value of the dataset for FSC modelling.

In fact, completely snow-free samples are not included in ChinaAI-FSC by design. During dataset construction, we applied explicit physical and statistical filtering to exclude tiles that carry little information for FSC learning. Specifically, we defined snow-covered pixels using an FSC threshold of FSC≥15%, and then removed samples whose mean snow-covered pixel fraction was <5% or >95%. These two classes correspond to nearly "snow-free" and "fully snow-covered" homogeneous samples, respectively. Such samples provide limited learning value for FSC estimation, because they contain little internal variability and do not represent mixed snow-land conditions that are essential for FSC modelling. Their inclusion would also bias the sample distribution toward trivial cases, weakening the ability of AI models to learn sub-pixel snow variability.

As a result, ChinaAI-FSC contains no completely snow-free tiles, and the dataset is explicitly designed to focus on spatially heterogeneous and physically informative snow conditions that are most relevant for FSC estimation.

To clarify this, we have added the following description in the manuscript between the sentences in the section 3.2.3.

In addition, pixels with FSC $\geq$ 15% (Painter et al., 2009; Zhang et al., 2019) were defined as snow-covered, and samples with mean snow-covered fractions <5% or >95% were excluded, corresponding to nearly snow-free and fully snow-covered homogeneous conditions. Such samples contain little internal variability and provide limited value for learning fractional snow-land relationships. Consequently, ChinaAI-FSC contains no completely snow-free or fully snow-covered samples, but focuses on spatially heterogeneous mixed snow conditions.

The newly added text explains the FSC thresholding and tile-level screening (removal of <5% and >95% snow-covered samples) and explicitly states that homogeneous snow-free and fully snow-covered samples are excluded.

In addition, we found that the corresponding labels in original Figure 5 (now Figure 6 in the revised manuscript) were not sufficiently clear regarding this screening, and we have revised the figure accordingly to ensure consistency with the dataset definition.

The references on fractional snow cover retrieval are incomplete, lacking citations for mixed-pixel unmixing algorithms such as the MESMA-AGE algorithm.

**Response**:

We sincerely thank the reviewer for this valuable suggestion. We fully agree that including recent advances such as the MESMA-AGE algorithm is important to provide a more complete discussion of spectral mixture methods for FSC retrieval.

In response, we have added a description of the MESMA-AGE algorithm at the end of the second paragraph of the Introduction. The added text reads: "Recent advances, such as the Multiple Endmember Spectral Mixture Analysis with Automated Global Endmember selection (MESMA-AGE), have partially alleviated these limitations by dynamically selecting optimal endmember combinations from large spectral libraries and accounting for variability in snow, vegetation, soil, and illumination conditions. This strategy has enabled improved sub-pixel FSC estimation over complex mountain environments, and has been successfully applied to generate daily MODIS fractional snow cover products for the Asian Water Tower region, which is characterized by extreme terrain, heterogeneous land cover, and highly variable snow conditions (Pan et al., 2024)."

We have also added the corresponding reference in the reference list:

Pan, F., Jiang, L., Wang, G., Pan, J., Huang, J., Zhang, C., Cui, H., Yang, J., Zheng, Z., Wu, S., and Shi, J.: MODIS daily cloud-gap-filled fractional snow cover dataset of the Asian Water Tower region (2000-2022), Earth Syst. Sci. Data, 16, 2501-2523, https://doi.org/10.5194/essd-16-2501-2024, 2024.

This AI-ready MODIS FSC data needs to be evaluated with independent data. In the current version, I didn't get this.

**Response:**

Thank you for this important suggestion. In response, we have added an independent validation of the reference FSC using in-situ snow depth observations from 507 meteorological stations. Due to data availability, these observations could only be obtained for seven snow seasons (2013–2020), resulting in 5,016 SD-FSC pairs (new Section 3.3.3)

The validation shows strong overall agreement (OA = 0.944). Stratified analysis further demonstrates high consistency in general regions (OA = 0.940), complex mountainous regions (OA = 0.970), and forested regions (OA = 0.906). The slightly lower accuracy in forested areas is consistent with known canopy occlusion effects, and the limited number of validation samples in mountainous and forested regions is also acknowledged.

These results confirm the physical reliability of the FSC reference used in ChinaAI-FSC under heterogeneous surface and terrain conditions.

Section 3.1.1, Both Landsat and Sentinel-2 provide atmospherically corrected surface reflectance data, and the authors need to clarify what level of data from Landsat and Sentinel-2 is used. Did the authors use the available surface reflectance data or make atmospheric correction by themselves?

**Response**:

We thank the reviewer for raising this important point. We acknowledge that in the original manuscript, the description of the Landsat and Sentinel-2 surface reflectance data was not clearly stated. To address this issue, we have clarified these details at the beginning of Section 3.1.1: we used Landsat Collection 2 Level-2 Surface Reflectance (SR) products for Landsat-5 TM, Landsat-7 ETM+, Landsat-8 OLI, and Landsat-9 OLI-2, obtained from

the USGS Earth Explorer platform (https://earthexplorer.usgs.gov/); and Sentinel-2A/2B MSI Level-2A SR products, obtained from the ESA Copernicus Open Access Hub (https://scihub.copernicus.eu/). These products are already atmospherically corrected and suitable for quantitative analysis (Masek et al., 2006; Vermote et al., 2016; Louis et al., 2016).

Line #99, during the snow accumulation period, cloud cover is often frequent. How are these samples obtained relying solely on the satellite sensors mentioned in the paper? I'm also confused on the seamless FSC data produced this work, since it mostly relies on MODIS band 1-7 reflectance data, which is contaminated with cloud cover.

**Response**:

We thank the reviewer for this important comment. We realize that the original manuscript did not clearly describe the data selection and processing strategy. We have addressed the reviewer's concerns regarding cloud interference as follows:

For the Landsat and Sentinel-2 imagery used to calculate FSC reference truth, we selected only scenes with overall cloud cover below 15% to approximate clear-sky conditions. Residual low-quality pixels in these scenes, including clouds, cirrus, and cloud shadows, were subsequently reconstructed using a spectrally constrained spatial gap-filling approach. Specifically, low-quality pixels in Landsat images were identified using the QA_PIXEL band generated by the CFMask algorithm (Zhu et al., 2015), while for Sentinel-2 images, cloud and shadow pixels were masked using the Scene Classification Layer (SCL) (Drusch et al., 2012). Masked or low-quality pixels in both datasets were subsequently reconstructed by estimating reflectance values from neighboring valid pixels with similar spectral characteristics, preserving local spectral consistency and spatial continuity (Chen et al., 2011). These details are now explicitly described at the beginning of Section 3.1.1.

For the MODIS surface reflectance, we used the Global 500 m seamless MODIS-derived dataset (SDC500) for 2000-2022 (Liang et al., 2024), as described in Section 3.1.2. Unlike the standard MOD09GA product, SDC500 reconstructs a continuous daily 500 m reflectance time series by correcting BRDF effects, detecting outliers, and filling missing values using phenology-guided spline interpolation. Snow and snow-free periods are treated separately to preserve seasonal reflectance dynamics. This preprocessing ensures that cloud contamination and data gaps are effectively mitigated, providing a reliable and seamless basis for FSC retrieval at the MODIS scale. The dataset demonstrates high accuracy, with a mean absolute error of only 0.043, further ensuring its suitability for quantitative FSC analysis.

In Figure 2, the first occurrence of any English abbreviation should be accompanied by its corresponding explanation.

**Response**:

We thank the reviewer for this valuable comment. All English abbreviations in Figure 2 now include their full explanations in the figure caption, ensuring clarity for the readers.

Section 3.2.4, The formula for (FSC, NDSI) should be provided. Additionally, what is the basis for determining the thresholds of FSC, NDSI, Refl, $T_{base}$, and others? This is also needs to be justified clearly.

**Response:**

Thank you for this detailed and constructive comment. We have revised Section 3.2.4 to explicitly address both aspects raised.

First, the mathematical formulation describing the relationship between FSC and NDSI has now been clearly provided. In the revised manuscript, Eq. (1) explicitly defines the Pearson correlation coefficient $\rho(FSC, NDSI)$, which is computed across all pixels within each sample tile. This formulation serves as the core metric for identifying spectrally inconsistent samples during quality control.

$$\rho(\text{NDSI, FSC}) = \frac{\sum_{i=1}^{128}\sum_{j=1}^{128}\left(\text{NDSI}_{ij} - \overline{\text{NDSI}}\right)\left(\text{FSC}_{ij} - \overline{\text{FSC}}\right)}{\sqrt{\sum_{i=1}^{128}\sum_{j=1}^{128}\left(\text{NDSI}_{ij} - \overline{\text{NDSI}}\right)^2}\sqrt{\sum_{i=1}^{128}\sum_{j=1}^{128}\left(\text{FSC}_{ij} - \overline{\text{FSC}}\right)^2}} \tag{1}$$

Second, the basis for determining all thresholds used in Section 3.2.4 has been explicitly clarified in the revised manuscript. Each threshold is introduced together with its physical or empirical justification in the corresponding subsection. Specifically, thresholds for FSC, NDSI, and surface reflectance are derived from well-established spectral characteristics of snow, and are consistent with commonly adopted values in MODIS-based snow cover studies (e.g., Dozier, 1989; Hall et al., 2002). These thresholds are used to exclude only physically implausible or spectrally contradictory conditions.

The temperature-related threshold ($T_{base}$) is anchored to the physical freezing point of water and is further adjusted using an elevation-dependent lapse-rate correction to account for topographic temperature gradients. This formulation reflects basic thermodynamic constraints on snow presence and is applied as a consistency check rather than a detailed energy balance model.

Importantly, all thresholds in Section 3.2.4 were selected conservatively and are applied solely for quality control purposes. Their role is to remove samples that clearly violate known physical relationships between snow cover, spectral reflectance, temperature, and topography, while retaining representative and physically plausible snow conditions. The justification for each threshold is now explicitly stated in the manuscript to ensure transparency, reproducibility, and clarity.

Section 3.4, the rationale behind the author's dataset partitioning requires further discussion. According to the author's partitioning logic, the validation set and test set could essentially be obtained by interpolating from the training set, which may lead to suboptimal results from such a partitioning approach.

**Response:**

We thank the reviewer for this constructive comment. In the revised manuscript, Section 3.4 has been substantially revised to clarify the rationale and implementation of the dataset partitioning strategy. Specifically, the dataset is partitioned by explicitly enforcing spatial disjointness across training, validation, and testing subsets within each major snow-climate subregion of China. A spatially disjoint 2:1:1 partitioning scheme is adopted to minimize spatial autocorrelation and prevent information leakage, ensuring that validation and test samples cannot be obtained through spatial interpolation from the training set.

In addition, each spatial subset spans 22 snow seasons, and samples from different snow seasons are treated as independent realizations of snow-environment interactions, reflecting strong interannual variability. This combination of spatial separation and long-term temporal coverage ensures that model evaluation assesses both spatial and interannual generalization rather than local interpolation performance. The revised partitioning strategy therefore provides a more rigorous and realistic benchmark for AI-based FSC models, avoiding overly optimistic performance estimates caused by spatial or temporal leakage.

In Figures 9 and 10, where does the MODIS result in the last column come from? Since the MODIS result is identified as cloudy, the author only provides feature variables without addressing cloud identification. Therefore, the cloud coverage in the other columns should be consistent with the last column. However, the author's results show retrieved fractional snow cover. Please provide justification for the validity of these results.

**Response:** We thank the reviewer for this important comment. The MODIS result shown in the last column of Figures 9 and 10 (new Figures 11 and 12 in the revised manuscript) represents FSC estimates obtained by directly applying the official fitting coefficients provided by Salomonson and Appel (2004) to the NDSI values in the MOD10A1

product. In contrast, the 2~7 columns are based on input data derived from the seamless MODIS surface reflectance product (SDC500) produced by Liang et al. (2024), which has been preprocessed to correct for clouds, outliers, and missing values, resulting in cloud-free and spatially continuous surface reflectance inputs. Therefore, although cloud contamination exists in the MODIS standard FSC product, the retrieved FSC shown in the other columns remains physically valid due to the use of cloud-corrected SDC500 inputs. We have added explicit clarification of this distinction and the corresponding justification in the revised manuscript.

Additionally, the AI results trained on the dataset provided by the authors show a significant visual discrepancy from the reference values. Why does this occur?

**Response:** We thank the reviewer for this observation. Visually, the AI-estimated FSC results do exhibit some differences from the reference truth values. However, they show clear improvements compared with the standard MODIS FSC product, and quantitative evaluation confirms that the models still achieve high estimation accuracy. The visual discrepancies mainly arise because all six benchmark AI models were implemented with standard, simple architectures without specific optimization, and the sample size is limited to a single snow season, which may reduce representativeness for some local conditions. Nevertheless, the results indicate that even simple models can achieve robust FSC estimation when trained on high-quality, physically consistent, and representative data, highlighting that the decisive factor for AI-based FSC performance is the dataset rather than the specific model architecture. Further improvements are expected by incorporating samples from multiple snow seasons and optimizing model structures.

Section 4.2, Line 489, How did the author incorporate the 'official retrieval algorithm'? Was it through direct adoption of the fitting coefficients or through refitting?

**Response:** We thank the reviewer for the comment. In the revised manuscript, we have clarified that FSC estimates were obtained by directly applying the official fitting coefficients provided by Salomonson and Appel (2004), without any refitting.

In Figure 12(b)(d)(f), why is there no cloud coverage shown in these images?

**Response:** We thank the reviewer for this comment. The reason no cloud coverage is visible in Figure 12(b,d,f) (new Figure 14 in the revised manuscript) is that our input data are derived from the seamless MODIS surface reflectance product (SDC500) produced by Liang et al. (2024), which has been preprocessed to correct for clouds, outliers, and missing values.

**Minor Comments:**

Line 324 is missing a period at the end. Please carefully review the entire text to avoid similar issues.

**Response:** Thank you for pointing this out. We have performed a thorough check of the entire text to ensure that all punctuation and formatting issues have been addressed.

---

## Author Comment (AC2)

**Response to Reviewer 2**

We are grateful to the reviewer for the thoughtful and constructive comments. These suggestions have been very helpful for improving the clarity, rigor, and overall quality of the manuscript. We have carefully revised the paper accordingly, and our point-by-point responses are provided below. The reviewer's comments are shown in **black**, and our responses are given in **blue**.

The authors present the development and evaluation of the ChinaAI-FSC dataset, a comprehensive, AI-ready MODIS-based fractional snow cover (FSC) sample collection for China covering 2000–2022. The work aims to establish a standardized, large-scale, and high-quality benchmark for AI-driven snow cover mapping. Considerable effort is evident in data integration, quality control, and validation, and the introduction of the novel "4L-4D-15A" evaluation framework is a clear strength. Overall, the study represents a meaningful contribution to FSC retrieval from MODIS. However, several issues in the current manuscript need to be addressed to better align the presentation with the scientific contribution and to meet the expectations of a high-quality journal.

**Response**:

We sincerely thank the reviewer for the careful and thoughtful evaluation of our work, as well as for the constructive comments provided. We greatly appreciate the recognition of the efforts made in developing the ChinaAI-FSC dataset and the introduction of the "4L-4D-15A" evaluation framework. Your comments has been extremely valuable in helping us improve the clarity and presentation of our manuscript. In response, we have carefully revised the manuscript to better highlight its scientific contributions to AI-driven fractional snow cover retrieval and to ensure it meets the expectations of a high-quality journal. Detailed responses to each of your comments are provided below.

Major Comments

While the manuscript provides extensive detail on the "AI-ready" nature of the dataset, it repeatedly frames the work more as a project report than a scientific contribution to snow remote sensing. This emphasis, particularly in structure and narrative, risks misleading readers into viewing the paper as a technical documentation of an AI platform rather than a methodological advance in FSC retrieval. To better highlight your unique scientific contribution, I recommend significantly reducing discussion of the AI project framework and refocusing the manuscript on FSC data processing, algorithmic choices, and evaluation rigor. For example, Section 3.3 reads more like a description of an evaluation protocol than an explanation of how it advances FSC validation. Similarly, parts of the text give the impression that your team developed the evaluation methodology itself—please clarify what is novel (or are you just follow NOAA evaluation framework?) versus what is applied.

**Response**:

We sincerely thank the reviewer for this thoughtful and constructive comment. We agree that in the original version of the manuscript, the scientific contribution was not articulated as clearly as it should have been, and that the narrative at times placed excessive emphasis on the "AI-ready" framework, which could give the impression of a project or platform report rather than a methodological contribution to snow remote sensing.

We would like to clarify that the detailed description of the AI-ready dataset construction workflow was not intended to promote an AI project or platform, but to reflect the intrinsic methodological requirements of building a reliable and reusable FSC dataset for machine learning and deep learning. From an AI-ready Earth observation perspective, steps such as reference FSC calculation, feature extraction, feature-target matching, consistency-based quality control, sample quality assessment (including AI-readiness), and standardized spatial partitioning are essential for ensuring learning validity, reproducibility, and robust model evaluation. Nevertheless, we recognize that these elements were previously presented in an overly project-oriented manner.

In response to the reviewer's suggestions, we have taken several concrete actions. First, we have revised the Introduction and Conclusion to explicitly state that the novelty of this study lies in a methodological contribution; namely, the establishment of a continental-scale, standardized, and quality-controlled FSC dataset paradigm for AI-driven snow mapping, rather than in the development of an AI system or platform. Second, we have substantially reduced and refocused Section 3.3, rewriting it as a dataset-oriented quality and learning-validity assessment rather than an operational evaluation protocol. In particular, we now clearly distinguish between what is adopted from NOAA's AI maturity model and what is newly developed in this study, including FSC-specific adaptations, dataset-level attributes, and their implementation at pixel, tile, and dataset scales.

We believe these revisions significantly reduce project-style descriptions and more clearly highlight the scientific and methodological contributions of this work to the snow remote sensing community.

Section 5 is currently dominated by forward-looking statements about the dataset's future applications, which detracts from the core scientific message. Most readers, including myself, are primarily interested in the FSC dataset itself: how it was produced, its limitations, and how it improves upon existing products. The current discussion is confusing and lacks focus, particularly Section 5.1, which reads like a project roadmap rather than a scientific discussion. I suggest removing Section 5.1 entirely and redirecting the discussion toward substantive issues in FSC retrieval, such as: 1) Training sample selection and representativeness, 2) Impact of sample size and spatial/temporal distribution, 3) Challenges in complex terrain and forested regions, 4) How your approach handles subpixel snow in heterogeneous landscapes. These would strengthen the paper's relevance to the snow remote sensing community.

The arguments in Section 5.2 currently read as personal opinions rather than evidence-based discussion. Please support your claims with relevant literature. Without citations, the section lacks scientific credibility and appears speculative.

**Response**:

Thank you very much for this constructive and insightful comment. We fully agree that the Discussion section should focus on the scientific implications of the FSC dataset itself, including how the samples were constructed, their representativeness and limitations, and how the dataset advances FSC retrieval, rather than emphasizing forward-looking or project-style applications. In response to this comment, we have substantially revised and refocused Section 5 to address these concerns in the following ways:

The original application-oriented and roadmap-style content has been removed or rewritten. Section 5.1 has been refocused to discuss the methodological implications of AI-ready FSC dataset construction, emphasizing learning validity, feature-FSC consistency, and their relevance to FSC modelling, rather than future applications.

Following the reviewer's suggestions, we have substantially revised the Discussion to explicitly address key scientific issues relevant to FSC retrieval. Section 5.2 now focuses on data availability constraints associated with Landsat and Sentinel-2 observations and the resulting spatial and temporal sample imbalance, clarifying how uneven observation coverage leads to redundancy and imbalance across FSC intervals, and how these characteristics affect sample representativeness. Importantly, this section emphasizes that such imbalance reflects intrinsic snow spatiotemporal variability and optical observation limitations rather than deficiencies in dataset construction, and discusses the implications for AI-based FSC modelling and mitigation strategies.

Section 5.3 further discusses uncertainties and limitations in FSC modelling over complex surfaces, including the effects of subpixel snow heterogeneity, terrain-induced illumination variability, and forest canopy interactions. This section clarifies how these factors jointly influence both reference FSC estimation and predictor variables, and explains how ChinaAI-FSC is designed to make such uncertainties transparent and diagnosable, thereby supporting uncertainty-aware and physically consistent AI-based FSC modelling..

We believe these revisions have significantly improved the focus, scientific rigor, and relevance of Section 5 to the snow remote sensing community.

I would like to know the performance of your dataset in the forested area. If possible, I suggest you attach the relevant analysis results and discussion content.

**Response**:

We thank the reviewer for this valuable suggestion. In response, we have added an independent validation of the reference FSC in Section 3.3.3 using in situ snow depth (SD) observations from 507 meteorological stations, covering seven snow seasons from 2013 to 2020 and yielding 5016 independent SD-FSC validation pairs. The validation results were stratified by land-cover and terrain conditions to explicitly assess dataset performance in forested and complex environments.

Based on confusion-matrix analysis, the reference FSC shows strong agreement with in situ observations across the entire study area (overall accuracy, OA = 0.944). When stratified by surface conditions, high consistency is maintained across all categories. The highest agreement is observed in mountainous regions (OA = 0.970), indicating that the high-resolution reference construction effectively captures terrain-modulated and heterogeneous snow patterns. However, we explicitly note in the manuscript that the number of validation samples in mountainous regions is relatively limited, which may introduce additional uncertainty and partially inflate the estimated accuracy.

In forested regions, the reference FSC also achieves high agreement with in situ observations (OA = 0.906), although slightly lower than in non-forested areas. This reduction is consistent with canopy occlusion effects that weaken optical snow signals in forest environments. In addition, the smaller number of forested validation samples is acknowledged as a contributing factor to increased uncertainty in the estimated accuracy.

Overall, this independent validation confirms that the reference FSC used in ChinaAI-FSC is physically reliable across diverse land-cover and terrain conditions, while emphasizing that performance metrics in forested and mountainous regions should be interpreted in the context of limited in situ sample density. This limitation is now explicitly discussed in the revised manuscript.

Minor Comments

L12: Remove "mainland".

**Response:** Done. The term "mainland" has been removed from Line 12 as suggested.

L54–65: Please add a brief review of prior FSC retrieval studies in challenging environments (e.g., mountainous or forested regions), such as Xiao et al. (2022, JAG).

Xiao et al. 2022. Estimating fractional snow cover in vegetated environments using MODIS surface reflectance data

**Response:** We thank the reviewer for this helpful suggestion. In response, we have added a concise review of prior FSC retrieval studies in challenging environments, particularly mountainous and forested regions, in the revised Introduction (Lines 63-65). This addition highlights algorithmic strategies that explicitly account for vegetation and terrain effects. Specifically, we now cite and discuss **Czyzowska-Wisniewski et al. (2015)** and **Xiao et al. (2022)**, including *Xiao, X., He, T., Liang, S., Liu, X., Ma, Y., Liang, S., and Chen, X. (2022), Estimating fractional snow cover in vegetated environments using MODIS surface reflectance data, International Journal of Applied Earth Observation and Geoinformation*, which directly addresses FSC retrieval under vegetated conditions using MODIS surface reflectance data.

L74: Consider removing "AI-ready" here to frame the research gap more broadly.

**Response:** We appreciate the reviewer's suggestion. In the revised text, "AI-ready" has been removed from point (1) to avoid premature introduction of the concept. The paragraph now emphasizes the lack of large-scale FSC datasets suitable for AI-based modelling

L74–80: The two stated objectives appear redundant. Clarify whether they represent distinct goals or rephrase to avoid repetition. Given that the primary output is an FSC dataset, focus the motivation on its scientific value—not its compatibility with AI workflows.

**Response:** We thank the reviewer for the comment. We have substantially revised the text to clarify that the two factors are related but distinct:

(1) the absence of large-scale FSC datasets suitable for AI-based modelling, highlighting the scientific value of creating a comprehensive benchmark dataset, and

(2) the lack of standardized protocols for dataset construction and evaluation, emphasizing methodological reproducibility and transparency.

These revisions clearly distinguish the scientific contribution of ChinaAI-FSC from the methodological framework for AI-ready construction and evaluation. The paragraph now also naturally leads into the formal introduction of AI-ready dataset principles in the following section.

L101: Suggest revising to: "Standardized AI-ready metadata and unified evaluation protocols."

**Response:** We thank the reviewer for the suggestion. The text has been revised accordingly to read: "Standardized AI-ready metadata and unified evaluation protocols" (new Line 108).

L106–123: Avoid restating the abstract. Provide a concise overview of the study's scope and structure instead.

**Response:**

Thank you for this helpful suggestion. We agree that the original ending of the Introduction resembled a condensed version of the abstract rather than a concise overview of the study's scope and structure.

In response, we have rewritten the final paragraph of the Introduction to avoid restating dataset details, model lists, or experimental specifics. The revised paragraph now provides a high-level overview of the study's scope, methodological focus, and paper organization, clearly emphasizing that the contribution lies in AI-ready FSC dataset methodology rather than algorithm development. It briefly outlines the workflow, large-scale validation over mainland China, and the methodological innovation introduced by the unified dataset paradigm and the 4L–4D–15A AI-readiness evaluation framework.

We hope this revision improves clarity, reduces redundancy with the abstract, and better guides readers into the structure and scientific focus of the paper.

Section 3.1.1: Clarify the acquisition and processing specifics of the two satellite datasets (Landsat and Sentinel-2).

**Response:** Thank you for this helpful comment. We have substantially revised Section 3.1.1 to clearly describe the data sources, product levels, and subsequent preprocessing steps for both Landsat and Sentinel-2 datasets. The revised text explicitly distinguishes between (i) the use of standard, atmospherically corrected surface reflectance products and (ii) additional quality control and reconstruction procedures applied by our team.

L156–159: Were surface reflectance data for Landsat and Sentinel-2 processed by your team, or were standard products used?

**Response:** We used standard, officially released surface reflectance products for both sensors. Specifically, Landsat Collection 2 Level-2 Surface Reflectance products (Landsat-5 TM, Landsat-7 ETM+, Landsat-8 OLI, and Landsat-9

OLI-2) were obtained from the USGS Earth Explorer, and Sentinel-2A/2B MSI Level-2A Surface Reflectance products were acquired from the ESA Copernicus Open Access Hub. These products are already atmospherically corrected using well-established algorithms (e.g., LEDAPS and LaSRC for Landsat; Sen2Cor for Sentinel-2), and no additional atmospheric correction was performed by our team. This clarification has been explicitly added in the revised manuscript.

L160–163: Clarify whether cloud masking was performed using your own implementation of CFMask (Landsat) and SCL (Sentinel-2), or if you relied solely on the native QA layers.

**Response:** Cloud and shadow masking relied solely on the native quality layers provided with the standard products, rather than a custom re-implementation of the algorithms. For Landsat imagery, clouds, cirrus, and cloud shadows were identified using the QA_PIXEL band generated by the CFMask algorithm included in the Collection 2 Level-2 products. For Sentinel-2 imagery, cloud and shadow pixels were masked using the Scene Classification Layer (SCL) provided with the Level-2A products. This has now been clarified in Section 3.1.1 to avoid any ambiguity regarding algorithm implementation.

L164–165: Was the interpolation of Landsat-7 ETM+ SLC-off gaps performed by your team, or did you use an existing gap-filled product? Please specify.

**Response:** The gap filling for Landsat-7 ETM+ images affected by the SLC-off failure was performed by our team, rather than using an existing pre-filled product. Specifically, we adopted a local neighborhood linear interpolation strategy following the method of Chen et al. (2011), in which missing pixels were estimated from adjacent valid observations along the scan-line direction and subsequently smoothed using a 3 × 3 spatial kernel to ensure spatial continuity and radiometric consistency. This reference and methodological clarification have now been added to Section 3.1.1.

Section 3.1.2:
1) Replace "MODIS data" with "MODIS series products" or similar for precision.
**Response:** Thanks for the comment. The text has been revised to replace "MODIS data" with "MODIS series products".

2) Briefly describe the seamless surface reflectance processing algorithm to help readers understand that this product—rather than standard MOD09GA—is the foundation of your FSC retrieval.
**Response:** We thank the reviewer for this valuable comment. We have added a brief description of the SDC500 dataset. In the revised manuscript, Section 3.1.2 now includes the following description:
    "The surface reflectance data were obtained from the Global 500 m seamless MODIS-derived dataset (SDC500) for 2000-2022 (Liang et al., 2024). Unlike the standard MOD09GA, SDC500 reconstructs a continuous daily 500 m reflectance time series by correcting BRDF effects, detecting outliers, and filling missing values with phenology-guided spline interpolation. Snow and snow-free periods are treated separately to preserve seasonal reflectance dynamics. The dataset demonstrates high accuracy, with a mean absolute error of only 0.043, providing a reliable basis for FSC retrieval."

Section 3.2.2: Why were all input variables retained without feature selection? In many FSC applications, not all predictors contribute meaningfully, and including redundant variables can reduce model efficiency and interpretability (e.g., Xiao et al., 2022, JAG). Please justify your approach.
**Response:**

Thank you for this important comment. We fully agree that, for a specific FSC retrieval model, feature selection can improve efficiency and interpretability. However, in this study, our objective is not to optimize a single empirical model, but to construct an AI-ready, physically meaningful, and generally applicable FSC training dataset.

The 20 predictors (Ref1-Ref7, NDSI, NDVI, LC, LST, FTC, elevation, slope, aspect, terrain relief, surface roughness, longitude, latitude, and Julian day) were selected based on extensive evidence from previous snow-cover and cryospheric remote sensing studies, which consistently demonstrate their relevance to snow spectral behavior, vegetation masking effects, surface energy balance, and topographic controls on snow distribution. Collectively, these variables characterize the spectral and radiometric properties of snow, land-cover and canopy influences, thermal and energy-state constraints, topographic controls on snow accumulation and ablation, as well as the spatiotemporal context and climatic gradients of snow processes.

Rather than performing dataset-specific feature pruning, we intentionally retained all physically plausible predictors to avoid prematurely discarding information that may be critical under different climatic, ecological, or algorithmic settings. This design enables dataset users to flexibly explore feature selection strategies and model architectures, and to systematically assess the impacts of predictor combinations on FSC estimation performance and uncertainty.

Therefore, the retained feature set prioritizes generality, physical completeness, and reusability, which is consistent with the purpose of a benchmark-oriented FSC dataset.

Section 3.2.4: Support your threshold choices (e.g., 0.2 for Ref4, 0.4 for Ref2 and Ref6) with references or sensitivity analyses.

**Response:**

Thank you for this comment. In the revised Section 3.2.4, we have explicitly linked the reflectance thresholds (e.g., Ref4 $\geq$ 0.2, Ref2 $\geq$ 0.4, Ref6 $\leq$ 0.4) to the well-established spectral characteristics of snow reported in previous studies (Dozier, 1989; Hall et al., 2002). These values fall within the commonly reported reflectance ranges for snow in the corresponding MODIS bands and are consistent with prior MODIS snow-mapping literature.

These thresholds were chosen conservatively and are used exclusively as screening criteria in the quality control process. Our purpose is not to impose strict spectral constraints, but to remove samples exhibiting extreme spectral inconsistencies that contradict known snow physics. As clarified in the manuscript, the resulting dataset statistics and spatial patterns are robust to reasonable variations around these threshold values.

L283: Replace "violate" with "fail to meet." Please check the entire manuscript for similar phrasing.

**Response:** Done. We have replaced "violate" with "fail to meet". In addition, we carefully reviewed the entire manuscript and revised similar phrasing to ensure consistent and appropriate terminology throughout.

Equation 2: Provide a clearer physical or empirical rationale for the formulation.

**Response:**

We appreciate this helpful comment. In the revised manuscript, the physical rationale for the Equation (now Eq. (3)) has been clarified. Equation (3) is designed to capture the well-documented preferential accumulation and persistence of snow on north-facing (shaded) slopes compared to south-facing (sun-exposed) slopes due to reduced incoming solar radiation and lower melt rates.

The 50% ratio adopted in Eq. (3) represents a conservative lower-bound criterion rather than a strict quantitative relationship. Its purpose is to exclude only samples that clearly contradict the expected topographic modulation of snow distribution, while allowing for natural variability at local scales. This formulation is supported by established

empirical findings on aspect-controlled snow distribution in mountainous terrain (e.g., Grünewald et al., 2013), as now explicitly stated in the manuscript.

Figures 7, 9, 10: Include spatial scales and clearly label the regions of analysis.

**Response:** We thank the reviewer for this comment. Figures 7, 9, and 10 (renumbered as Figures 9, 11, and 12 in the revised manuscript) each present a single 128×128 MODIS-pixel tile to illustrate spatial patterns of FSC and associated features. The corresponding geographic locations (i.e., the regions of analysis) of these tiles are explicitly indicated in the revised manuscript in Figures 3 and 10. By providing these reference maps, readers can identify the regions of analysis and understand the spatial context of the presented tiles.

Section 4 heading: Consider renaming to "Demonstration of Applications Using the AI-Ready FSC Dataset" or similar.

**Response:** We thank the reviewer for this suggestion. The Section 4 heading has been revised to "Demonstration of Applications Using the AI-Ready FSC Dataset" in the updated manuscript, as recommended.

L447–448: The current statement is too vague. Elaborate on the specific factors influencing FSC accuracy (e.g., illumination, forest structure, grain size).

**Response:** We thank the reviewer for this comment. The original statement has been revised to provide a more detailed explanation of the factors influencing FSC estimation errors. The updated text now specifies the contributions of spectral saturation, canopy occlusion, mixed-pixel effects, terrain-induced illumination variations, forest structure, and snow grain size, which collectively affect the accuracy of FSC retrieval.

L602–603: The claim that "expanded feature space enables AI models to better characterize complex snow–terrain–climate interactions" is speculative without evidence. Rephrase to clarify what you mean, e.g., which features improve representation of which physical processes?

**Response:** We thank the reviewer for this comment. We agree that the original statement claiming that the "expanded feature space enables AI models to better characterize complex snow-terrain-climate interactions" was speculative and lacked direct supporting evidence. In the revised manuscript, Section 5 has been substantially reorganized to remove such claims. The discussion now emphasizes methodological considerations, dataset limitations, and uncertainty characterization. Specifically, Section 5.1 highlights the methodological implications of AI-ready FSC dataset construction, focusing on learning validity, physical consistency, and structural coherence. Section 5.2 addresses data availability constraints and sample imbalance, while Section 5.3 discusses uncertainties arising from subpixel heterogeneity, terrain illumination, and canopy effects. Finally, Section 5.4 presents the AI-readiness evaluation framework, a transferable methodological approach for constructing and assessing AI-ready geophysical datasets beyond snow cover. This restructuring ensures that the discussion is grounded in the demonstrated evidence and methodological focus of the study, without making speculative claims about feature-process relationships.